# Prolyl dihydroxylation of unassembled uS12/Rps23 regulates fungal hypoxic adaptation

Sara J Clasen, Wei Shao, He Gu, Peter J Espenshade*

Department of Cell Biology, Johns Hopkins University School of Medicine, Baltimore, United States

**Abstract** The prolyl-3,4-dihydroxylase Ofd1 and nuclear import adaptor Nro1 regulate the hypoxic response in fission yeast by controlling activity of the sterol regulatory element-binding protein transcription factor Sre1. Here, we identify an extra-ribosomal function for uS12/Rps23 central to this regulatory system. Nro1 binds Rps23, and Ofd1 dihydroxylates Rps23 P62 in complex with Nro1. Concurrently, Nro1 imports Rps23 into the nucleus for assembly into 40S ribosomes. Low oxygen inhibits Ofd1 hydroxylase activity and stabilizes the Ofd1-Rps23-Nro1 complex, thereby sequestering Ofd1 from binding Sre1, which is then free to activate hypoxic gene expression. In vitro studies demonstrate that Ofd1 directly binds Rps23, Nro1, and Sre1 through a consensus binding sequence. Interestingly, Rps23 expression modulates Sre1 activity by changing the Rps23 substrate pool available to Ofd1. To date, oxygen is the only known signal to Sre1, but additional nutrient signals may tune the hypoxic response through control of unassembled Rps23 or Ofd1 activity.

DOI: https://doi.org/10.7554/eLife.28563.001

## Introduction

Eukaryotic cells require oxygen and must adapt to changes in its supply to maintain homeostasis. Research on mechanisms underlying oxygen homeostasis has focused mainly on hypoxia-inducible factor (HIF) signaling in mammals (*Semenza, 2012*). HIFs are heterodimeric transcription factors whose activity is regulated by molecular oxygen through the prolyl hydroxylase domain (PHD) enzymes. These enzymes hydroxylate HIF proline residues to promote HIF degradation using 2-oxo-glutarate (2OG) and oxygen as co-substrates and Fe(II) as a co-factor (*Kaelin and Ratcliffe, 2008*). The PHDs belong to a large family of non-heme, 2OG/Fe(II)-dependent oxygenases whose members catalyze a broad set of reactions in eukaryotes and bacteria (*Hausinger, 2015*). Despite this diversity, the active site of most 2OG oxygenases is located in a highly-conserved double-stranded beta-helix fold with the Fe(II) coordinated by a catalytic triad (HxD/E...H) (*McDonough et al., 2010*). Since oxygen is a co-substrate for this enzyme family, they can act as sensors of cellular oxygen supply, making them ideal regulators for pathways requiring oxygen (*Ratcliffe, 2013*).

Lipid synthesis is highly oxygen-consumptive with yeast ergosterol synthesis requiring 12 molecules of dioxygen (*Espenshade and Hughes, 2007*). As a result, oxygen supply and sterol synthesis are coupled through the transcription factor Sre1 in the fission yeast *Schizosaccharomyces pombe*. Sre1 is a sterol regulatory element-binding protein (SREBP) homolog required for adaptation to low oxygen environments (*Hughes et al., 2005*). Notably, Sre1 is conserved across fungi, and homologs in pathogenic fungi are required for virulence since the mammalian host environment is hypoxic (*Chang et al., 2007*; *Willger et al., 2008*; *Bien and Espenshade, 2010*). In *S. pombe*, Sre1 is synthesized as an ER membrane-bound precursor that is cleaved in the Golgi to release its N-terminal transcription factor domain (Sre1N) in response to sterol depletion during hypoxia (*Hughes et al., 2005*;

*For correspondence:
peter.espenshade@jhmi.edu

**Competing interests:** The authors declare that no competing interests exist.

**eLife digest** Animals, plants, and fungi need oxygen to release energy within their cells and for other chemical reactions. Enzymes that use oxygen typically become less active when less oxygen is available, and this makes them well suited to help cells sense oxygen. These enzymes include oxygenases, some of which modify proteins by adding oxygen to specific sites in a reaction called hydroxylation. Oxygenases control how mammals adapt to low levels of oxygen – a condition referred to as hypoxia. These enzymes achieve this by hydroxylating a protein – specifically a transcription factor – that turns on genes for survival in low oxygen. Cells quickly destroy the hydroxylated transcription factor but when oxygen is limiting, it remains unmodified. This means that, rather than being destroyed, the transcription factor binds DNA, and activates genes that keep the cells alive and growing in low oxygen.

In fission yeast, an oxygenase called Ofd1 controls the activity of a transcription factor called Sre1. Yeast requires Sre1 to grow when oxygen is limiting. Exactly how Ofd1 regulates Sre1 is unknown, but the mechanism is different from that in mammals because regulation of gene expression does not need Sre1 to be hydroxylated.

Now, Clasen et al. report that Ofd1 actually hydroxylates another protein called Rps23. This protein is one of about 80 that form the cell's protein-building machinery, the ribosome. It turns out that, before Rps23 becomes part of the ribosome, it binds Ofd1 in a complex with other proteins. The multi-protein complex then acts to hydroxylate and transport Rps23 into the nucleus, where ribosomes are built and where the cell stores its DNA. When little oxygen is around, Ofd1 cannot hydroxylate Rps23. This stops the complex from falling apart and traps Ofd1 away from the transcription factor Sre1. When not bound by Ofd1, Sre1 is free to turn on genes that allow growth at low levels of oxygen. Finally, Clasen et al. show that more unassembled Rps23 means less Ofd1 is available to inhibit Sre1, which controls the yeast cell's response to hypoxia.

Humans have proteins similar to Ofd1 and Rps23. As such, this pathway for sensing oxygen in yeast may occur in humans too. Further work is now needed to explore if other enzymes that hydroxylate ribosomal proteins work in a similar way.

DOI: https://doi.org/10.7554/eLife.28563.002

*Porter et al., 2010*; *Stewart et al., 2011*). Sre1N upregulates genes required for hypoxic growth and its own transcription. Upon reoxygenation, Sre1N signaling is rapidly down-regulated by the 2OG oxygenase Ofd1 (*Hughes and Espenshade, 2008*). Ofd1 interacts with Sre1N in the presence of oxygen to inhibit DNA-binding and accelerate degradation (*Lee et al., 2011*; *Porter et al., 2012*). In contrast to HIF regulation by PHDs, oxygen regulation of Sre1N does not require its hydroxylation by Ofd1. Rather, under hypoxia Ofd1 preferentially binds the nuclear import adaptor Nro1 (*Lee et al., 2009*). Binding to Nro1 prevents Ofd1 binding to Sre1N, allowing Sre1N to activate hypoxic gene expression. This oxygen-regulated switch in Ofd1 binding requires oxygenase activity. However, the mechanism by which oxygen elicits this change in binding remains unclear as does the precise role for Ofd1 enzyme activity in regulating Sre1N.

Previous work in budding yeast identified a role for the Ofd1 and Nro1 homologs Tpa1 and Ett1, respectively, in translation termination. These studies found that ribosomes from cells lacking Tpa1 read through stop codons with increased frequency relative to wild-type cells (*Keeling et al., 2006*). While Ofd1 and Nro1 function in opposition in the context of Sre1N regulation, *ett1Δ* cells share the same ribosomal read-through defect as *tpa1Δ* cells (*Henri et al., 2010*; *Rispal et al., 2011*). Importantly, Tpa1 oxygenase activity is required to rescue this phenotype, suggesting that the ribosome read-through defect is mediated by a Tpa1 enzyme substrate. More recently, several studies identified the small ribosomal protein uS12 as an enzyme substrate for Ofd1, Tpa1, and the homologs in *human* (OGFOD1) and fly (Sudestrada1) (*Loenarz et al., 2014*; *Singleton et al., 2014*; *Katz et al., 2014*). uS12 is an essential and universal ribosomal protein that functions in translation fidelity (*Sharma et al., 2007*). While the human and fly enzymes catalyze prolyl-3-hydroxylation of uS12 P62, the fungal homologs catalyze 3,4-dihydroxylation of P62. Prolyl-3-hydroxylation was implicated in translation fidelity, but the function of uS12 dihydroxylation in fungi is unknown. The finding that uS12 is an Ofd1 substrate categorizes Ofd1 as a ribosomal oxygenase (ROX) (*Ge et al., 2012*). This

small but growing enzyme family modifies ribosomal proteins from the large and small subunit in both eukaryotes and prokaryotes. For many ROXs, the function of the hydroxylated product remains unknown, similar to dihydroxylated uS12 P62 in fungi.

Here, we independently identified uS12, known as Rps23 in *S. pombe*, as a binding partner and substrate of Ofd1 and investigated the role that P62 dihydroxylation plays in fission yeast. We confirmed that Ofd1 dihydroxylates Rps23 and further report that Ofd1 and Rps23 form a complex with Nro1. This complex functions to transport newly synthesized Rps23 to the nucleus while simultaneously facilitating the dihydroxylation of P62. In addition, Ofd1 activity regulates formation of the Ofd1-Rps23-Nro1 complex such that decreased enzyme activity under low oxygen stabilizes the complex. As a result, unassembled Rps23 regulates Sre1N signaling by sequestering Ofd1 in an oxygen-dependent manner, thereby coupling hypoxic gene expression to rates of ribosomal synthesis. Finally, we identified a conserved Ofd1 binding sequence shared by all known Ofd1 binding partners. This study outlines a new paradigm for control of hypoxic adaptation, assigns a second function to uS12 hydroxylation, and defines a distinct mechanism by which an oxygenase functions as a cellular oxygen sensor.

## Results

### Ofd1 binds and dihydroxylates Rps23 P62

To identify conserved enzyme substrates of Ofd1/OGFOD1, we performed a yeast two-hybrid screen for human OGFOD1 binding partners using a human fetal brain cDNA library as prey. We isolated uS12 (coded by the *RPS23* gene) as a binding partner of OGFOD1 and confirmed that the interaction is conserved between Ofd1 and Rps23 in fission yeast (*Figure 1A*; *Figure 1—figure supplement 1A*).

To investigate whether Ofd1 binds directly to Rps23, we used an in vitro GST pull-down assay. In contrast to free GST, full-length GST-Rps23 bound Ofd1 (*Figure 1B*, lanes 2–3), indicating direct binding between Ofd1 and Rps23. Next, we mapped the region of Rps23 required for binding to Ofd1. Rps23 contains an evolutionarily divergent N-terminal extension domain (aa 1–46) followed by a conserved C-terminal globular domain (aa 47–143) (*Smith et al., 2008*). We fused GST to Rps23 truncations and found that Ofd1 binds to the N-terminal extension domain (*Figure 1B*, lanes 4–5). Furthermore, Rps23 aa 1–23 were necessary and sufficient to capture Ofd1 (*Figure 1B*, lanes 6–7). This region of Rps23 (*Figure 1C*, colored magenta) is buried in assembled 40S subunits and thus inaccessible to Ofd1 based on crystal structures of budding yeast ribosomes (*Ben-Shem et al., 2011*). Therefore, Ofd1 binds to Rps23 prior to its assembly into the 40S subunit in the nucleus. Since unassembled Rps23 represents only 2% of total Rps23 in cells (*Figure 1—figure supplement 1–1B*, lanes 1 and 4), we used ribosome-depleted lysates to test if Ofd1 binds Rps23 in vivo. Immunopurified wild-type Ofd1 bound unassembled Rps23 in the presence of the crosslinker DSP, but failed to bind Rps5, a ribosomal protein of similar size and charge (*Figure 1D*, lanes 7–8). In addition, the Fe(II)-binding mutant Ofd1 H142A D144A was unable to pull down Rps23 (*Figure 1D*, lanes 11–12), indicating that an intact Ofd1 active site is required to bind Rps23.

Since oxygenase function is required for Ofd1-Rps23 binding, we hypothesized that Rps23 is a substrate of Ofd1. We identified Rps23 P62 as a promising target for hydroxylation based on earlier work that reported a +16 Dalton (Da) mass shift for mammalian RPS23 aa 61–68: QPNSAIRK (*Louie et al., 1996*). P62 is essential and part of a highly conserved loop that projects into the decoding center of the ribosome upon anticodon binding (*Sharma et al., 2007*). We confirmed that the +16 Da mass shift occurred on this proline residue by incubating Ofd1 with purified 6xHis-RPS23 and performing MS/MS that targeted the QPNSAIR peptide (*Figure 1—figure supplement 1–1C*). In light of subsequent studies showing that Ofd1 and its fungal homologs dihydroxylate P62 (*Loenarz et al., 2014*), we developed a stable isotope labeling with amino acids in cell culture (SILAC) assay to quantify Rps23 P62 hydroxylation. Wild-type cells were labeled with heavy (H) lysine and *ofd1Δ* cells with light (L) lysine, mixed, and processed to enrich for Rps23 from assembled 40S subunits. Following digestion by LysC, H/L peptide ratios were measured by LC-MS/MS (*Figure 1E*). We detected only dihydroxylated P62 in wild-type cells, consistent with published findings (*Loenarz et al., 2014*). Analysis of the wild-type-*ofd1Δ* SILAC pair revealed a complete loss of dihydroxylated P62 in *ofd1Δ* cells, accompanied by the appearance of unmodified P62 (*Figure 1F*). We

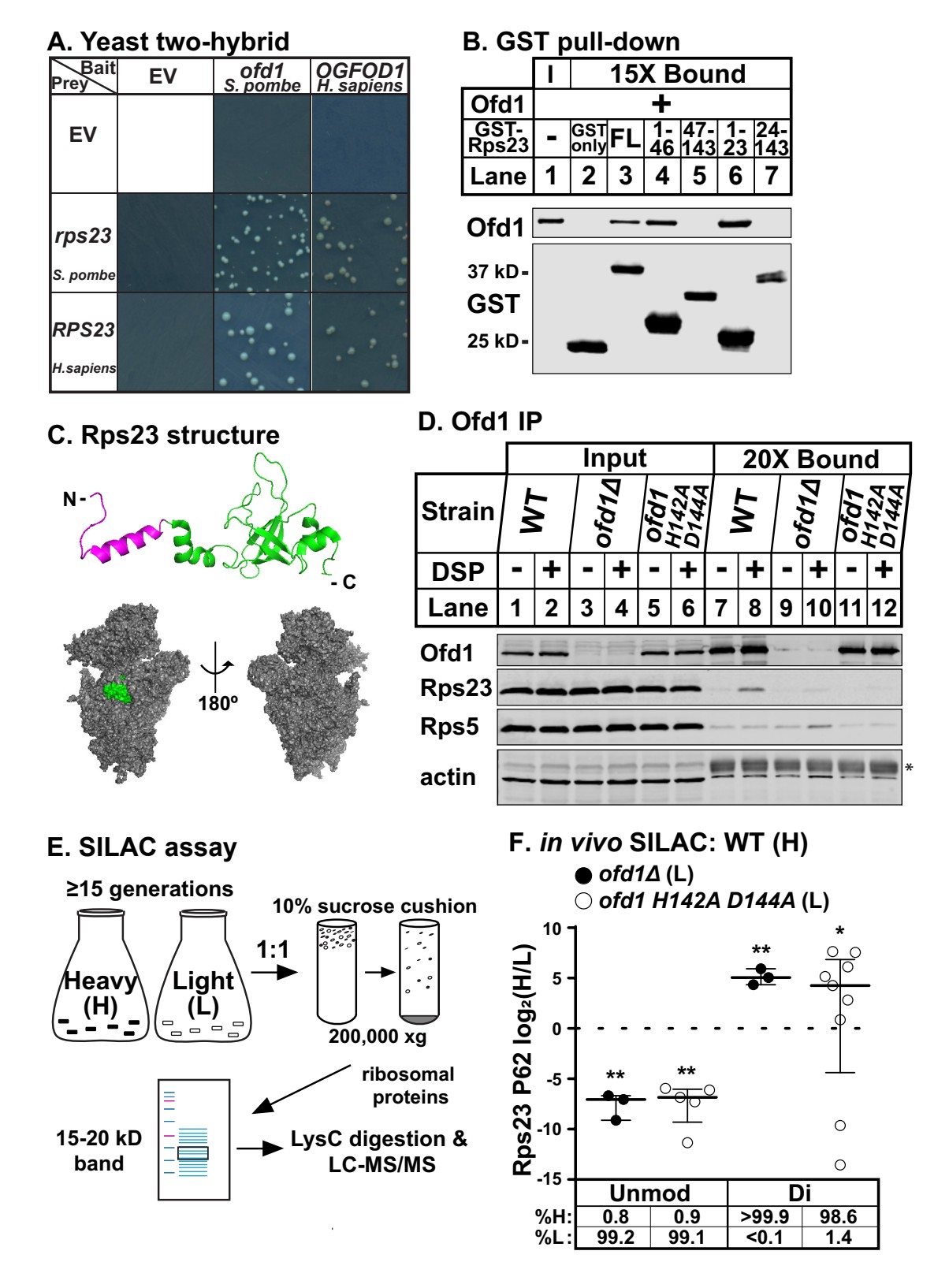

**Figure 1.** Ofd1 binds and dihydroxylates Rps23 P62. (A) Yeast two-hybrid assays were performed using *S. cerevisiae* AH109 strain transformed with indicated bait and prey vectors, either empty vector (EV) or expressing specified genes. Cells were grown on SD-Leu-Ade-Trp-His medium

*Figure 1 continued on next page*

Figure 1 continued

supplemented with X-α-Gal for 8 days. Shown are cropped regions of plates; see *Figure 1—figure supplement 1A* for uncropped plates. (B) Full-length (FL) and truncated Rps23 were N-terminally tagged with GST and purified on glutathione beads from bacterial lysates. Following incubation with purified Ofd1 (84.5 nM), bound proteins were eluted with reduced glutathione. Input and bound fractions were analyzed by immunoblot using antibodies against Ofd1 and GST. Shown are representative blots from one of three independent experiments. (C) *Top*: crystal structure of Rps23 in the *S. cerevisiae* ribosome (PDB 4V88) with the Ofd1-binding site, Rps23 (aa 1–23), colored magenta; *bottom*: surface representation of Rps23 in the 40S subunit (*left*: interface; *right*: solvent-exposed). (D) Wild-type, *ofd1Δ*, and *ofd1 H142A D144A* cells were treated with vehicle (8% DMSO) or crosslinker (2 mM DSP) for 5 min, lysed in detergent, and centrifuged to pellet ribosomes. Ofd1 was purified from ribosome-depleted lysates with anti-Ofd1 antibody, and input and bound fractions were analyzed by immunoblot with indicated antibodies. Asterisk (*) denotes IgG heavy chain (see *Figure 1—figure supplements 1–2* for additional information). Shown are representative blots from one of three independent experiments. (E) Diagram outlining SILAC assay to quantify Rps23 hydroxylation. Yeast cells were cultured in SILAC medium supplemented with either heavy (H) or light (L) lysine for ≥ 15 generations. Whole cell lysates were mixed 1:1 H:L and centrifuged through a sucrose cushion to collect ribosomes. Ribosomal proteins were extracted from pellets and separated by gel electrophoresis prior to LysC digestion and LC-MS/MS. (F) Wild-type, *ofd1Δ*, and *ofd1 H142A D144A* cells were cultured and processed as described in (E) with wild-type cells labeled with heavy lysine. Error bars represent the interquartile range of H/L ratios containing either unmodified (unmod) or dihydroxylated (di) Rps23 P62, and significance was tested by Mann-Whitney (*p<0.01; **p<0.0001; n.s.). Median %H and %L values are reported below. Shown are PSMs from one of four independent experiments. See also *Figure 1—source data 1*.
DOI: https://doi.org/10.7554/eLife.28563.003

The following source data and figure supplements are available for figure 1:

**Source data 1.** SILAC mass spectrometry data.
DOI: https://doi.org/10.7554/eLife.28563.006
**Figure supplement 1.** Ofd1 binds and dihydroxylates Rps23 P62.
DOI: https://doi.org/10.7554/eLife.28563.004
**Figure supplement 2.** Wild-type, *ofd1Δ*, and *ofd1 H142A D144A* cells were treated and processed as described in *Figure 1D*.
DOI: https://doi.org/10.7554/eLife.28563.005

found similar results for the SILAC pair of wild-type and enzyme-dead *ofd1 H142A D144A* (*Figure 1F*). Collectively, these data show that Ofd1 binds unassembled Rps23 (aa 1–23) and independently demonstrate that Ofd1 catalyzes Rps23 dihydroxylation.

## Identification of an Ofd1 binding sequence

While Rps23 is the only known enzyme substrate of Ofd1, we previously showed that Ofd1 directly binds Nro1, a negative regulator of Ofd1 and a nuclear import adaptor (*Lee et al., 2009*; *Yeh et al., 2011*). To determine if a common Ofd1 binding site exists, we aligned the sequences of Rps23 (aa 1–23) and Nro1 (aa 1–30) required for binding to Ofd1. We identified a stretch of identical and chemically similar residues (*Figure 2A*) and mutated each amino acid to aspartate in the context of either GST-Rps23 FL (*Figure 2B*) or GST-Nro1 aa 1–30 (*Figure 2C*) to test if these residues are required for binding to Ofd1. Mutating the N-terminal basic residue, proline, glycine, or leucine completely abolished binding to Ofd1 for both Rps23 and Nro1, while mutating a non-conserved residue adjacent to the proline had an intermediate effect (*Figure 2B–C*, lanes 4–8). Downstream mutations revealed slight differences between the Ofd1 binding sequences: mutations in conserved alanine residues resulted in a severe reduction in Rps23-Ofd1 binding, while the corresponding mutations in Nro1 only partially disrupted binding (*Figure 2B–C*, lanes 10–11). For both Rps23 and Nro1 though, mutations in the most C-terminal amino acids displayed only mild defects in Ofd1 binding relative to upstream residues (*Figure 2B–C*, lanes 13–15). These similarities between the Ofd1 binding sites in Rps23 and Nro1 argue for a conserved Ofd1 binding sequence.

## Ofd1, Rps23, and Nro1 form a complex

Given that both Rps23 and Nro1 contain an Ofd1 binding site, we hypothesized that these proteins compete for binding to Ofd1 and predicted that down-regulation or deletion of one binding partner would increase binding between Ofd1 and the other partner. While Nro1 is not essential (*Lee et al., 2009*), Rps23 is required for cell viability due to its role in translation (*Sharma et al., 2007*). Two genes – *rps23+* and *rps2302+* – code for identical Rps23 proteins. While deletion of either gene results in a modest decrease in total Rps23 levels, levels of unassembled Rps23 are dramatically reduced by ~90% (*Figure 1—figure supplement 1B*). *rps23Δ* and *rps2302Δ* cells thus represent severe knock-downs with respect to this Ofd1 binding partner. To test if Nro1 and unassembled Rps23 compete for binding to Ofd1, we immunopurified Ofd1 from wild-type, *nro1Δ*, *rps23Δ*, and

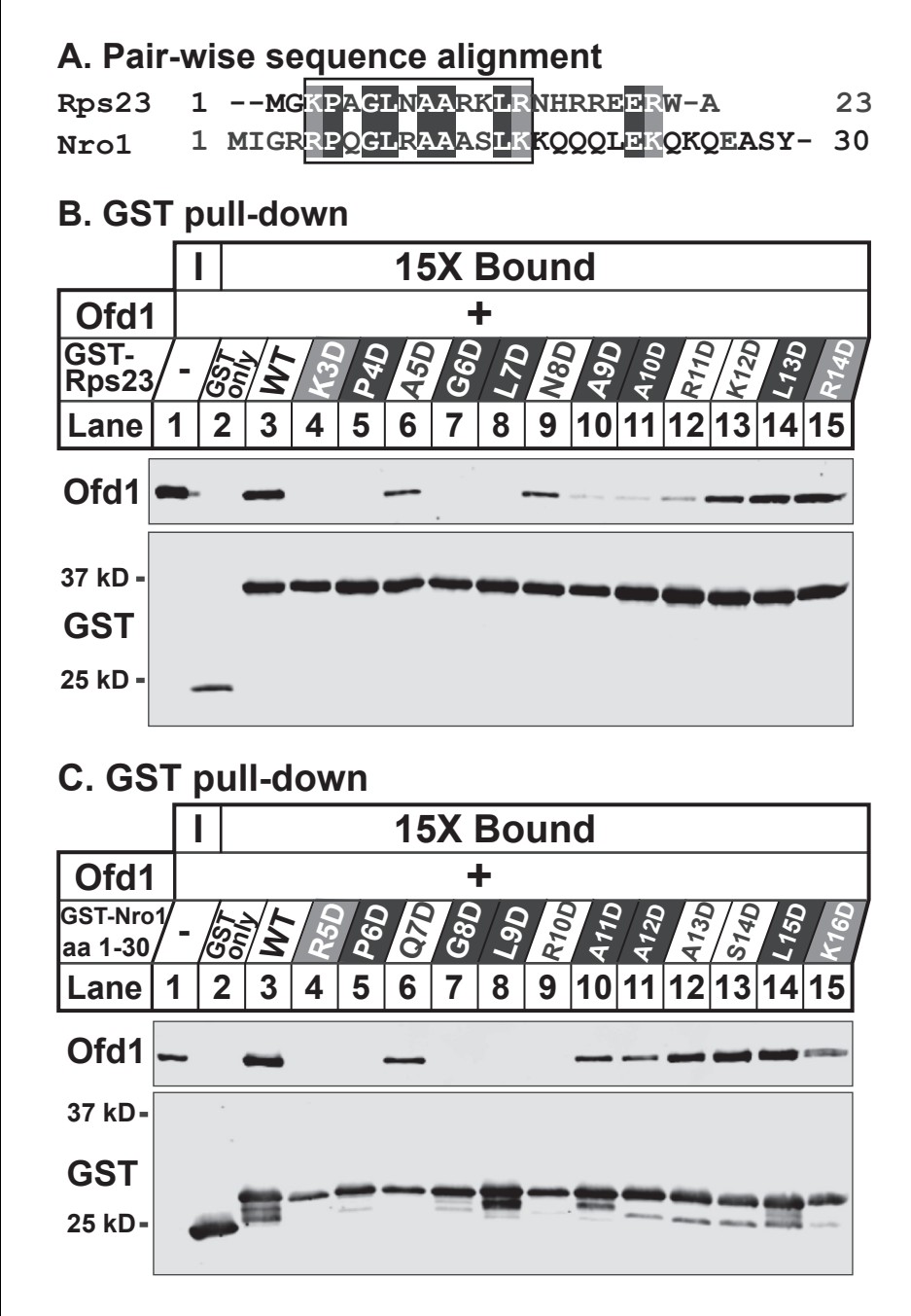

**Figure 2.** Identification of an Ofd1 binding sequence. (**A**) Pair-wise alignment of Rps23 (aa 1–23) and Nro1 (aa 1–30) sequences required to bind Ofd1; black boxes denote identical residues and gray boxes denote similar residues. Amino acids enclosed by the box were individually mutated and tested for binding to Ofd1. (**B, C**) Lysates of bacteria expressing wild-type or mutant GST-Rps23 FL (**B**) or GST-Nro1 aa 1–30 (**C**) were incubated with glutathione beads prior to incubation with purified Ofd1 (84.5 nM). Input and bound fractions were analyzed by immunoblot using antibodies against Ofd1 and GST. Shown are representative blots from one of at least two independent experiments.

DOI: https://doi.org/10.7554/eLife.28563.007

rps2302Δ lysates depleted of ribosomes and analyzed the bound fraction for Nro1 and Rps23. In wild-type cells treated with crosslinker, Ofd1 bound both Rps23 and Nro1, but not Rps5 (*Figure 3A*, lane 6). Surprisingly, Ofd1 failed to pull down Rps23 in *nro1Δ* cells (*Figure 3A*, lane 8), indicating that Ofd1-Rps23 binding requires Nro1. In addition, Ofd1 binding to Nro1 was weakened in *rps23Δ* and *rps2302Δ* cells (*Figure 3A*, lanes 9–10). In the absence of crosslinker, immunopurified Ofd1 bound less Nro1 in wild-type cells and the Ofd1-Rps23 interaction was barely detected (*Figure 3B*, lane 6). However, Ofd1-Nro1 binding was reduced in *rps23Δ* and *rps2302Δ* cells relative to wild-type cells across both conditions (*Figure 3A–B*; lanes 6, 9–10). Consistent with this, Nro1 bound less

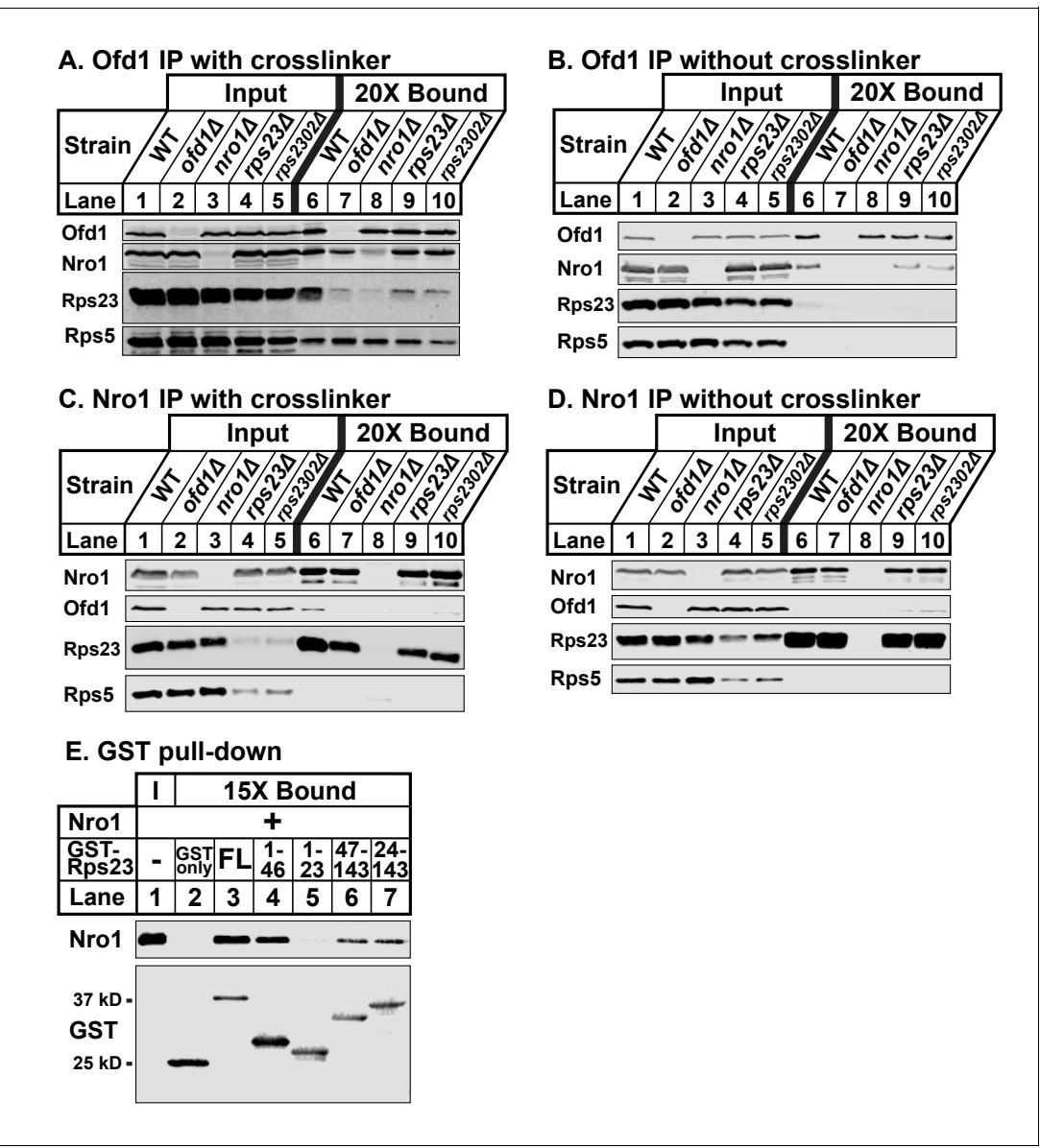

**Figure 3.** Ofd1, Rps23, and Nro1 form a complex. (A, B) Wild-type, *ofd1Δ*, *nro1Δ*, *rps23Δ*, and *rps2302Δ* cells were treated with crosslinker (A) (2 mM DSP) or without crosslinker (B), lysed in detergent, and depleted of ribosomes. Ofd1 was purified from lysates using anti-Ofd1 antibody, and input and bound fractions were analyzed by immunoblot with indicated antibodies. (C, D) Wild-type, *nro1Δ*, *ofd1Δ*, *rps23Δ*, and *rps2302Δ* cells were treated in the presence of crosslinker (C) or absence (D), lysed in detergent, and depleted of ribosomes. Nro1 was isolated from lysates using anti-Nro1 antibody, and input and bound fractions were analyzed by immunoblot with indicated antibodies. (E) GST-tagged full length (FL) and truncated Rps23 were purified on glutathione beads and incubated with purified Nro1 (84.5 nM). Input and bound fractions were analyzed by immunoblot using antibodies against Nro1 and GST. Representative blots from one of two independent experiments are shown for all figures.

DOI: https://doi.org/10.7554/eLife.28563.008

Ofd1 in *rps23Δ* cells and, to a lesser extent, *rps2302Δ* cells relative to wild-type when immunopurified from crosslinked samples (*Figure 3C*, lanes 6, 9–10). Unexpectedly, Nro1 showed strong binding to Rps23 in wild-type, *ofd1Δ*, *rps23Δ*, and *rps2302Δ* cells, independent of crosslinker (*Figure 3C–D*, lanes 6–7 and 9–10). Nro1-Rps23 binding was specific to Rps23 since Nro1 did not pull down Rps5 (*Figure 3C–D*, lanes 6–10). We consistently observed reduced unassembled Rps5 in *rps23Δ* and *rps2302Δ* cells, indicating that mechanisms exist to coordinate ribosomal protein expression. These results show that Rps23- and Nro1-binding to Ofd1 is interdependent rather than competitive, suggesting that they form a complex with Ofd1.

To test this further, we assayed Nro1 and Rps23 binding in vitro using a GST pull-down assay. Full-length GST-Rps23 directly bound purified Nro1 (*Figure 3E*, lane 3). Using Rps23 truncations, we then found that Rps23 contains at least two distinct binding sites for Nro1 because both GST-Rps23 aa 1–46 and GST-Rps23 aa 47–143 bound Nro1 (*Figure 3E*, lanes 4 and 6). However, the Ofd1 and Nro1 binding sites on Rps23 do not overlap as GST-Rps23 aa 1–23 failed to bind Nro1 (*Figure 3E*, lane 5). These in vitro binding studies, together with the immunopurifications, demonstrate that Ofd1, Rps23, and Nro1 form a complex in cells.

## Dihydroxylation of Rps23 P62 requires Nro1

The discovery that Nro1 is required for Ofd1 to bind to Rps23 in vivo (*Figure 3A*) led us to hypothesize that Nro1 functions in Rps23 hydroxylation. To test this, we examined Rps23 P62-containing peptides from wild-type and *nro1Δ* cells using SILAC. Interestingly, less than 10% of the detected dihydroxylated P62 signal originated from *nro1Δ* cells (*Figure 4A*), indicating that only 10% of the Rps23 is dihydroxylated in the absence of Nro1 compared to 100% in wild-type cells. Furthermore, we detected both unmodified and monohydroxylated P62 peptides with low H/L ratios, indicating that these peptides originated from *nro1Δ* cells (*Figure 4A*). To determine the relative abundance of the unmodified and monohydroxylated forms in cells lacking Nro1, we first measured the relative levels of unmodified Rps23 in *nro1Δ* and *ofd1Δ* cells using SILAC. We found that *nro1Δ* cells contributed 25% of the unmodified P62 signal, meaning that 32% of Rps23 P62 is unmodified in *nro1Δ* ribosomes. Consequently, monohydroxylated Rps23 P62 represents ~60% of the total Rps23 in these cells. (*Figure 4B*).

The substantial fraction of monohydroxylated P62 in *nro1Δ* cells shows that the efficient dihydroxylation of Rps23 P62 requires Nro1. We tested if Nro1 participates directly in the reaction by reconstituting the reaction in vitro. Purified Rps23 and Ofd1 were incubated alone or in the presence of purified Nro1 and then analyzed by Tandem Mass Tag (TMT)-labeled LC-MS/MS to quantify P62 hydroxylation. The ratio of Ofd1:Rps23:Nro1 (1:10:10) replicated the physiological ratio of Ofd1: unassembled Rps23:Nro1 in cells (data not shown). Consistent with the in vivo results, addition of Nro1 to the in vitro reaction increased Rps23 P62 dihydroxylation by 1.6 fold (*Figure 4C*). The increase in dihydroxylated Rps23 was accompanied by a corresponding decrease in monohydroxylated Rps23, suggesting a role for Nro1 in facilitating the second hydroxylation event. The presence of Nro1 also resulted in a small but significant reduction in the levels of the unmodified P62 substrate compared to the reaction lacking Nro1. Together, these in vivo and in vitro results demonstrate that complete Rps23 dihydroxylation requires Nro1.

## Nro1 imports Rps23 in complex with Ofd1

Thus far, our data shows that Ofd1 dihydroxylates unassembled Rps23 in a complex with Nro1. Since Nro1 is structurally similar to karyopherins and functions as a nuclear import adaptor for Ofd1 (*Yeh et al., 2011*), we tested if Nro1 plays a role in Rps23 localization. Wild-type cells that express Rps23-GFP from the *rps2302* locus showed Rps23-GFP localized to the nucleus by live-cell imaging (*Figure 5A*). The Rps23-GFP fusion protein does not assemble into functional 40S subunits (data not shown) and localized to the nucleus rather than being exported to the cytosol. In *nro1Δ* cells, Rps23-GFP was excluded from the nucleus despite only a minor reduction in Rps23-GFP expression (*Figure 5B*), indicating that Nro1 is required for nuclear import of unassembled Rps23. However, alternate pathways for Rps23 import must exist since *nro1Δ* cells are viable. Furthermore, Ofd1-dependent hydroxylation of Rps23 P62 is not required for nuclear import of Rps23. In both *ofd1Δ* and *ofd1 H142A D144A* cells, Rps23-GFP localizes to the nucleus, as does Rps23 P62A-GFP in

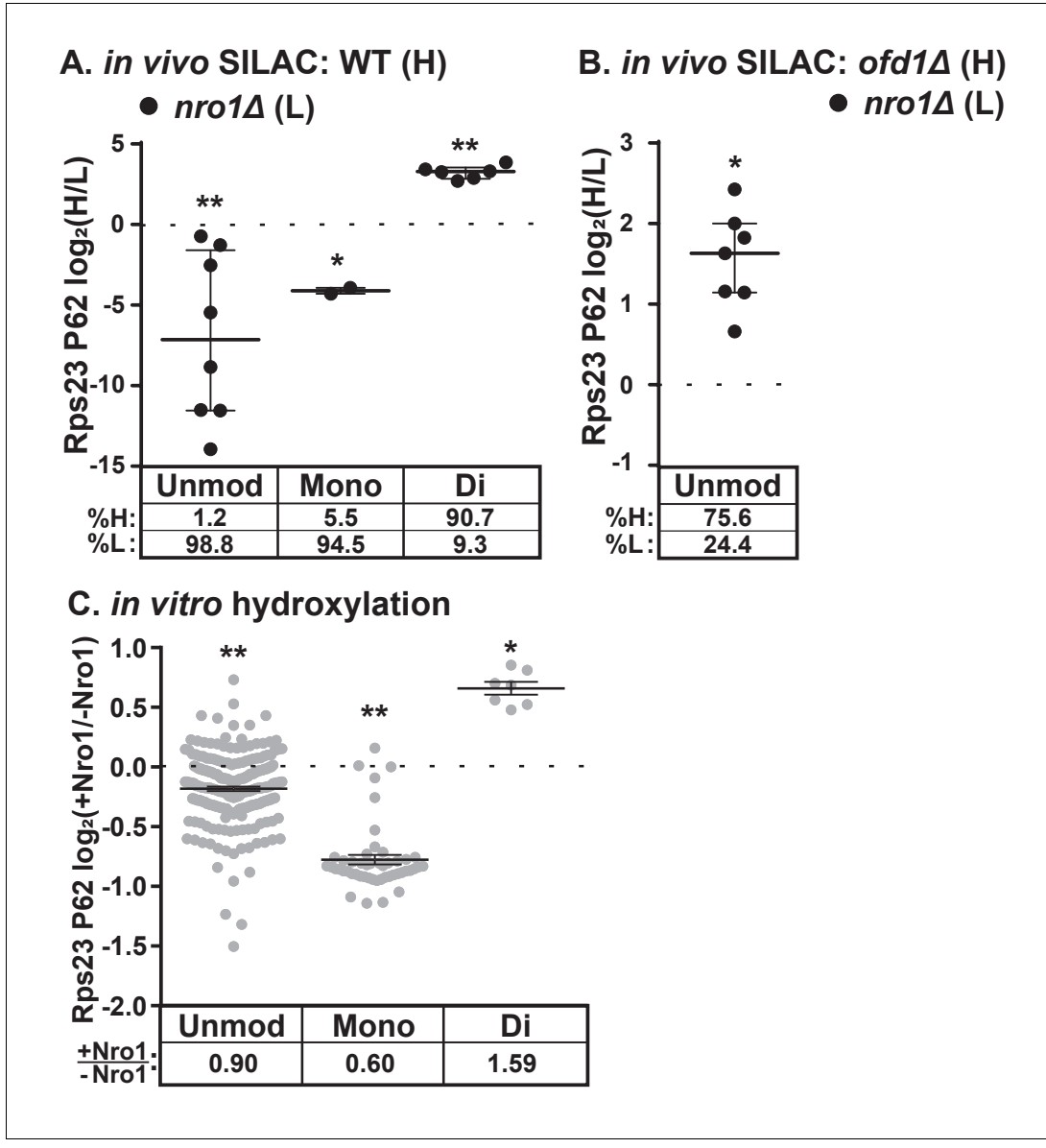

**Figure 4.** Dihydroxylation of Rps23 P62 requires Nro1. (**A**) Wild-type and *nro1Δ* cells were cultured and processed as described in *Figure 1E* with wild-type cells labeled with heavy lysine. Error bars represent the interquartile range and significance was determined by Mann-Whitney (*p<0.001, **p<0.0001). Median % heavy and % light are reported below. Shown are PSMs from one of four independent experiments. (**B**) *ofd1Δ* and *nro1Δ* cells were cultured and processed as described in *Figure 1E* with *ofd1Δ* cells labeled with heavy lysine to determine the relative amount of unmodified Rps23 P62 in *nro1Δ* cells. Mann-Whitney test, *p<0.0001. Median % heavy and % light are reported below. Shown are PSMs from one of two independent experiments. See also *Figure 1—source data 1*. (**C**) Ofd1 (0.5 μM) and MBP-Rps23 (5 μM) were incubated in the absence and presence of Nro1 (5 μM) as described in Methods for 1 hr followed by quantification of Rps23 P62 hydroxylation by TMT LC-MS/MS. Error bars represent ± SEM. Significance was calculated by Wilcoxon signed rank test (*p<0.05; **p<0.0001). Average fold changes (+Nro1/-Nro1) are reported below. Shown are PSMs from one of two independent experiments. See also *Figure 4—source data 1*.

DOI: https://doi.org/10.7554/eLife.28563.009

The following source data is available for figure 4:

**Source data 1.** TMT mass spectrometry data.

DOI: https://doi.org/10.7554/eLife.28563.010

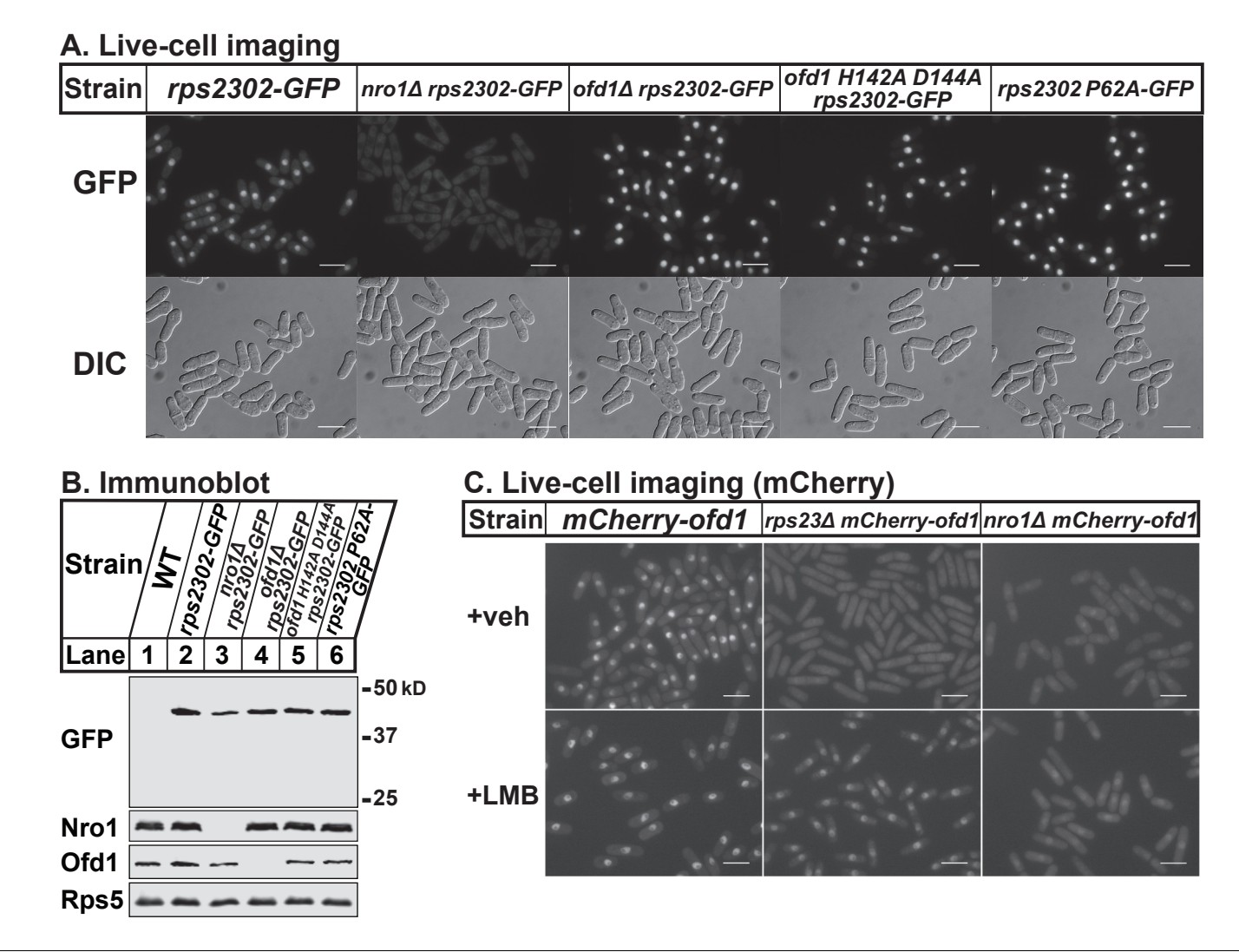

**Figure 5.** Nro1 imports Rps23. (**A**) Indicated strains were cultured in rich medium and imaged by fluorescence and differential interference contrast (DIC) microscopy. Scale bar = 10 microns. Shown are representative images from one of at least two independent experiments. (**B**) Whole cell extracts (50 μg) of indicated strains were analyzed by immunoblot with antibodies against GFP, Nro1, Ofd1, and Rps5. Shown are representative blots from one of at least two independent experiments. (**C**) *mCherry-ofd1*, *rps23Δ mCherry-ofd1*, and *nro1Δ mCherry-ofd1* cells were treated with vehicle (0.05% [v/v] ethanol) or the nuclear export inhibitor LMB (92.5 nM) for 1 hr prior to live-cell imaging by fluorescence microscopy. Scale bar = 10 microns. Shown are representative images from one of three independent experiments.

DOI: https://doi.org/10.7554/eLife.28563.011

*ofd1⁺* cells. (*Figure 5A*). These data indicate that Rps23 enters the nucleus regardless of hydroxylation state, and suggests that nuclear import of Rps23 is not blocked under hypoxic conditions.

Because both Rps23 and Ofd1 require Nro1 for nuclear localization, we hypothesized that Nro1 imports Rps23 and Ofd1 together as a complex. Therefore, we predicted that Ofd1 might be mislocalized in *rps23Δ* cells due to the dramatic reduction in unassembled Rps23 and consequent reduction in complex formation (*Figure 3A–D*). mCherry-Ofd1 expressed from the endogenous locus localized primarily to the nucleus in wild-type cells, consistent with published data (*Figure 5C*) (*Hughes and Espenshade, 2008*). Deletion of *rps23⁺* resulted in loss of nuclear localization with Ofd1 mislocalized throughout the cell (*Figure 5C*). This supports our hypothesis that Ofd1 and Rps23 traffic together to the nucleus with Nro1. Inhibition of Crm1-dependent nuclear export by leptomycin B (LMB) in *rps23Δ* cells restored Ofd1 nuclear localization, indicating that Ofd1 still traffics to the nucleus in these cells but at a reduced rate (*Figure 5C*). However, LMB treatment failed to

restore nuclear localization of mCherry-Ofd1 in *nro1Δ* cells, consistent with the requirement of Nro1 for Ofd1 nuclear import (*Figure 5C*) (*Yeh et al., 2011*). Collectively, these data suggest that unassembled Rps23 is imported into the nucleus in a complex with Nro1 and Ofd1 and that Rps23 dihydroxylation likely occurs in conjunction with nuclear import.

## Unassembled Rps23 regulates Ofd1 inhibition of Sre1N

In addition to their roles in Rps23 hydroxylation and nuclear import, Ofd1 and Nro1 are key regulators of Sre1 signaling and hypoxic adaptation in fission yeast (*Hughes and Espenshade, 2008*; *Lee et al., 2009*). Ofd1 inhibits signaling by binding Sre1N, which both prevents Sre1N DNA-binding and accelerates its degradation (*Lee et al., 2011*). Under low oxygen, Ofd1 binds Nro1, freeing Sre1N to activate gene expression. Given that Ofd1-Nro1 binding requires Rps23 (*Figure 3A–D*), we hypothesized that unassembled Rps23 functions as a positive regulator of Sre1N like Nro1.

To test if Rps23 regulates Sre1, we first analyzed growth of *rps23Δ* and *rps2302Δ* cells in the presence of the hypoxia-mimetic cobalt. In fission yeast, cobalt inhibits ergosterol synthesis, resulting in reduced ergosterol levels and the accumulation of methylated sterol intermediates (*Lee et al., 2007*). Cobalt likely disrupts ergosterol synthesis by causing defects in Fe-dependent enzymes since overexpression of *erg25+*, an Sre1N target gene and Fe-dependent enzyme, rescues growth on cobalt. Consequently, Sre1 activity is required for growth in the presence of cobalt and *sre1Δ* cells show severe growth defects on plates supplemented with cobalt chloride (*Stewart et al., 2011*). Growth on cobalt thus reflects the cell's capacity to activate Sre1. Like *nro1Δ* cells, *rps23Δ* and *rps2302Δ* cells failed to grow on cobalt chloride (*Figure 6A*). Consistent with this, *rps23Δ* and *rps2302Δ* cells, like *nro1Δ* cells, showed decreased Sre1N activation and reduced precursor levels under low oxygen (*Figure 6B*, lanes 7–12). In contrast, *ofd1Δ* cells resemble wild-type cells with respect to Sre1N levels despite the role of Ofd1 as a negative regulator of Sre1N (*Figure 6B*, lanes 1–2 and 5–6). Ofd1-independent mechanisms act upstream of Ofd1 to regulate the transport and cleavage of the Sre1 precursor to generate Sre1N (*Hughes and Espenshade, 2008*; *Stewart et al., 2011*). The failure of *rps23Δ* and *rps2302Δ* cells to grow on cobalt was due to defects in Sre1N production rather than general defects in translation since expression of Sre1N from a plasmid rescued growth on cobalt (*Figure 6C*). Furthermore, *rps23Δ* and *rps2302Δ* cells failed to upregulate the Sre1N target gene *hem13+* in the absence of oxygen, but hypoxic activation of the Sre1N-independent gene *erg1+* was normal (*Figure 6—figure supplement 6–1A*). Finally, the *rps23* mutant was the only ribosomal gene identified in both fission yeast deletion collection screens for cobalt-sensitivity (*Ryuko et al., 2012*; *Burr et al., 2016*), and we confirmed the specificity of the cobalt phenotype (*Figure 6—figure supplement 6–1B*). These two screens failed to identify *rps2302Δ* cells as cobalt-sensitive; however, we were unable to verify the deletion of *rps2302+* by PCR in the deletion collection strain (data not shown). These data demonstrate that Rps23 is a specific positive regulator of Sre1 signaling.

To test whether Rps23 regulates Sre1 after proteolytic activation, we used strains that express only soluble Sre1N from the *sre1* locus, bypassing the requirement for cleavage (*Hughes and Espenshade, 2008*). Low oxygen stabilized Sre1N in wild-type cells (*Figure 6D*, lanes 1–2), but *rps23Δ* and *rps2302Δ* cells failed to accumulate Sre1N under low oxygen (*Figure 6D*, lanes 4 and 6). Deletion of *ofd1+* in *rps23Δ* and *rps2302Δ* cells restored Sre1N expression to wild-type levels under low oxygen (*Figure 6D*, lanes 10 and 12), indicating that the defects in Sre1N accumulation are the result of increased inhibition by Ofd1 and further demonstrating that *rps23Δ* and *rps2302Δ* cells are not defective in Sre1N translation.

The model for Ofd1 inhibition of Sre1N predicts that Ofd1 acts on Sre1N directly, but direct binding has not been demonstrated and the Ofd1 binding site on Sre1N has not been identified. Previous yeast two-hybrid studies mapped the Ofd1 binding region to Sre1 aa 271–340 (*Lee et al., 2011*). Alignment of this region with the Ofd1 binding sites in Rps23 and Nro1 identified Sre1 aa 286–297 as a putative Ofd1 binding site (*Figure 6E*). To test if Sre1N binds Ofd1 directly, we performed an in vitro binding assay and found that GST-Sre1 aa 271–340 bound purified Ofd1 (*Figure 6F*, lane 3). Mutational analysis showed that Sre1N-Ofd1 binding required Sre1N residues 286–293 and 297 (*Figure 6F*, lanes 4–11 and 15). In contrast, individually mutating Sre1N aa 294–296 impaired Ofd1 binding but did not abolish the interaction (*Figure 6F*, lanes 12–14). These data indicate that Sre1N binds Ofd1 directly through a conserved motif shared with Rps23 and Nro1.

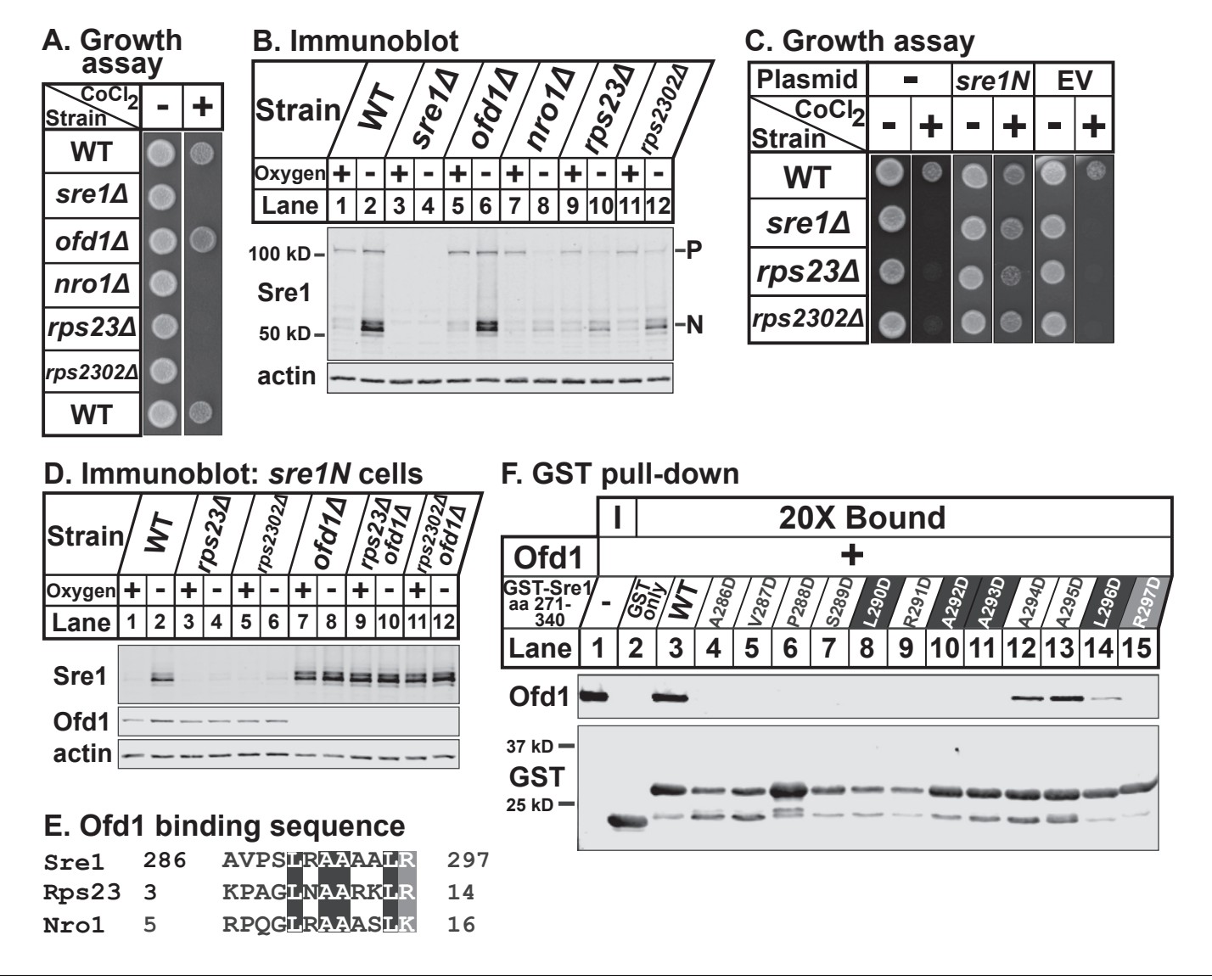

**Figure 6.** Unassembled Rps23 regulates Ofd1 inhibition of Sre1N. (A) Indicated strains were grown on rich medium in the absence and presence of 1.6 mM cobalt chloride and imaged after 3 and 4 days, respectively (5000 cells per spot). Shown are representative images from one of two independent experiments. (B) Indicated strains were cultured in rich medium in the presence or absence of oxygen for 3 hr. Whole cell extracts (20 μg) were treated with alkaline phosphatase and analyzed by immunoblot with antibodies against Sre1 and actin. P=precursor; N=nuclear. Shown are representative blots from one of two independent experiments. (C) Wild-type, sre1Δ, rps23Δ, and rps2302Δ cells were transformed with EV or CaMV-sre1N expressing vector and grown on rich medium for 3 days or rich medium containing 1.6 mM cobalt chloride (CoCl2) for 6 days (5000 cells per spot). Shown are representative images from one of two independent experiments. (D) Indicated sre1N strains were cultured in rich medium in the presence or absence of oxygen for 3 hr. Whole cell extracts (20 μg) were phosphatase-treated and analyzed by immunoblot with antibodies against Sre1, Ofd1, and actin. Shown are representative blots from one of two independent experiments. (E) Multiple sequence alignment of Sre1 aa 286–297, Rps23 aa 3–14, and Nro1 aa 5–16. (F) Lysates from bacteria expressing wild-type or mutant GST-Sre1 aa 271–340 were incubated with glutathione beads and then with purified Ofd1 (84.5 nM). Input and bound fractions were analyzed by immunoblot using antibodies against Ofd1 and GST. Shown are representative blots from one of two independent experiments.

DOI: https://doi.org/10.7554/eLife.28563.012

The following figure supplement is available for figure 6:

**Figure supplement 1.** Unassembled Rps23 regulates Ofd1 inhibition of Sre1N.

DOI: https://doi.org/10.7554/eLife.28563.013

Collectively, these experiments demonstrate that unassembled Rps23 functions with Nro1 as a positive regulator of Sre1N activity by binding Ofd1 and preventing its binding to Sre1N.

## Ofd1-Rps23-Nro1 complex sequesters Ofd1 under hypoxia to activate Sre1N

Our data indicate that Ofd1 binds independently to both a complex of Rps23-Nro1 and Sre1N, a dimeric, basic helix-loop-helix leucine zipper transcription factor (*Párraga et al., 1998*). Given that these proteins share a consensus Ofd1 binding sequence, Ofd1 likely forms a dimer, which is consistent with structural studies (*Figure 7A*) (*Kim et al., 2010*; *Henri et al., 2010*; *Horita et al., 2015*). Deletion of *nro1+* reduces Ofd1-Rps23-Nro1 complex formation (*Figure 3A*) and inhibits Sre1N signaling (*Lee et al., 2009*), indicating that Rps23-Nro1 and Sre1N dimer compete for binding to Ofd1. Finally, Rps23 P62 hydroxylation by Ofd1 requires oxygen (*Loenarz et al., 2014*). Based on these data, we proposed a model in which Ofd1 is free to bind Sre1N and inhibit transcriptional activity under normoxia. Under hypoxia, Ofd1 hydroxylation of Rps23 is inhibited, sequestering Ofd1 in complex with Rps23-Nro1 and freeing Sre1N to activate hypoxic gene expression (*Figure 7A*). Consistent with this, a 90% reduction in unassembled Rps23 (*rps23Δ* cells; *Figure 1—figure supplement 1–1B*, lanes 4–5) resulted in impaired Sre1N signaling under low oxygen due to increased inhibition by Ofd1 (*Figure 6D*). *rps23Δ* cells fail to produce sufficient Rps23 to effectively sequester Ofd1 under hypoxia.

Based on this model, we hypothesized that elevated levels of unassembled Rps23 would further sequester Ofd1 and stabilize Sre1N in the presence of oxygen. We overexpressed Rps23 in *2XSRE-ura4+* reporter cells and analyzed growth on plates lacking uracil (*Figure 7B*) (*Lee et al., 2009*). In this reporter strain, *sre1N* and *sre1Δ* cells failed to grow in the absence of uracil when transformed with the empty vector control (*Figure 7B*, spots 1 and 3). However, Rps23 overexpression in *sre1N* cells, but not *sre1Δ* cells, supported growth in the absence of uracil, indicating activation of Sre1 signaling (*Figure 7B*, spots 2 and 4). Furthermore, overexpression of GST-Rps23 in *sre1N* cells led to the accumulation of Sre1N under normoxia compared to control cells (*Figure 7C*). GST-Rps23 serves as a proxy for unassembled Rps23 given its ability to bind both Ofd1 and Nro1 (*Figures 1B* and *3E*) while being unable to assemble into functional 40S subunits: cells that express GST-Rps23 from the *rps2302* locus are not viable in an *rps23Δ* background (*Figure 7—figure supplement 1A*). These data support the sequestration model by demonstrating that changes in Rps23 levels affect Sre1 signaling.

Next, we tested if the Ofd1-binding sites in Nro1 and Rps23 mediate the sequestration of Ofd1. We mutated the Ofd1-binding site at several conserved residues in Nro1 and Rps23 and expressed these mutants in *nro1Δ sre1N* and *rps2302Δ sre1N* cells, respectively. These cells carrying empty vector failed to grow in the presence of cobalt (*Figure 7D*), consistent with the fact that both *nro1Δ sre1N* and *rps2302Δ sre1N* cells were unable to activate Sre1N under hypoxia due to Ofd1-dependent inhibition (*Lee et al., 2009*; *Figure 6D*, lanes 5–6, 11–12). We predicted that expression of wild-type Nro1 and Rps23 would rescue growth on cobalt by sequestering Ofd1, but that Nro1 and Rps23 mutants that fail to bind Ofd1 would consequently fail to restore growth.

*nro1Δ sre1N* cells expressing wild-type Nro1 grew on cobalt while cells expressing the Ofd1-binding mutants - Nro1 P6D, G8D, and A11D – phenocopied empty vector controls (*Figure 7D*). Consistent with these cobalt growth defects, cells expressing Nro1 P6D, G8D, and A11D failed to activate Sre1N under low oxygen despite equal Nro1 expression levels (*Figure 7—figure supplement 7–1B*, lanes 3–10). Expression of Nro1 L15D, a mutant predicted to retain Ofd1-binding (*Figure 2*), partially rescued both growth on cobalt and Sre1N accumulation under hypoxia (*Figure 7D*; *Figure 7—figure supplement 1B*, lanes 4,12). *rps2302Δ sre1N* cells expressing wild-type *rps2302+* grew on cobalt (*Figure 7D*). Unexpectedly, although Rps23 P4D, G6D, and A9D are defective for Ofd1-binding in vitro (*Figure 2B*), *rps2302Δ sre1N* cells expressing these mutants showed partial to full recovery on cobalt relative to cells expressing wild-type Rps23 or Rps23 L13D, a mutant that binds Ofd1 in vitro. Furthermore, the Rps23 mutants except for G6D restored Sre1N activity under low oxygen (*Figure 7—figure supplement 1C*). Reduced expression of Rps23 G6D may explain its inability to fully rescue Sre1N levels (*Figure 7—figure supplement 1D*). These findings indicate that although the Ofd1-binding sites in Nro1 and Rps23 behave similarly in vitro, these sequences do not contribute equally to Ofd1 sequestration in vivo, with sequestration more dependent on Nro1 residues.

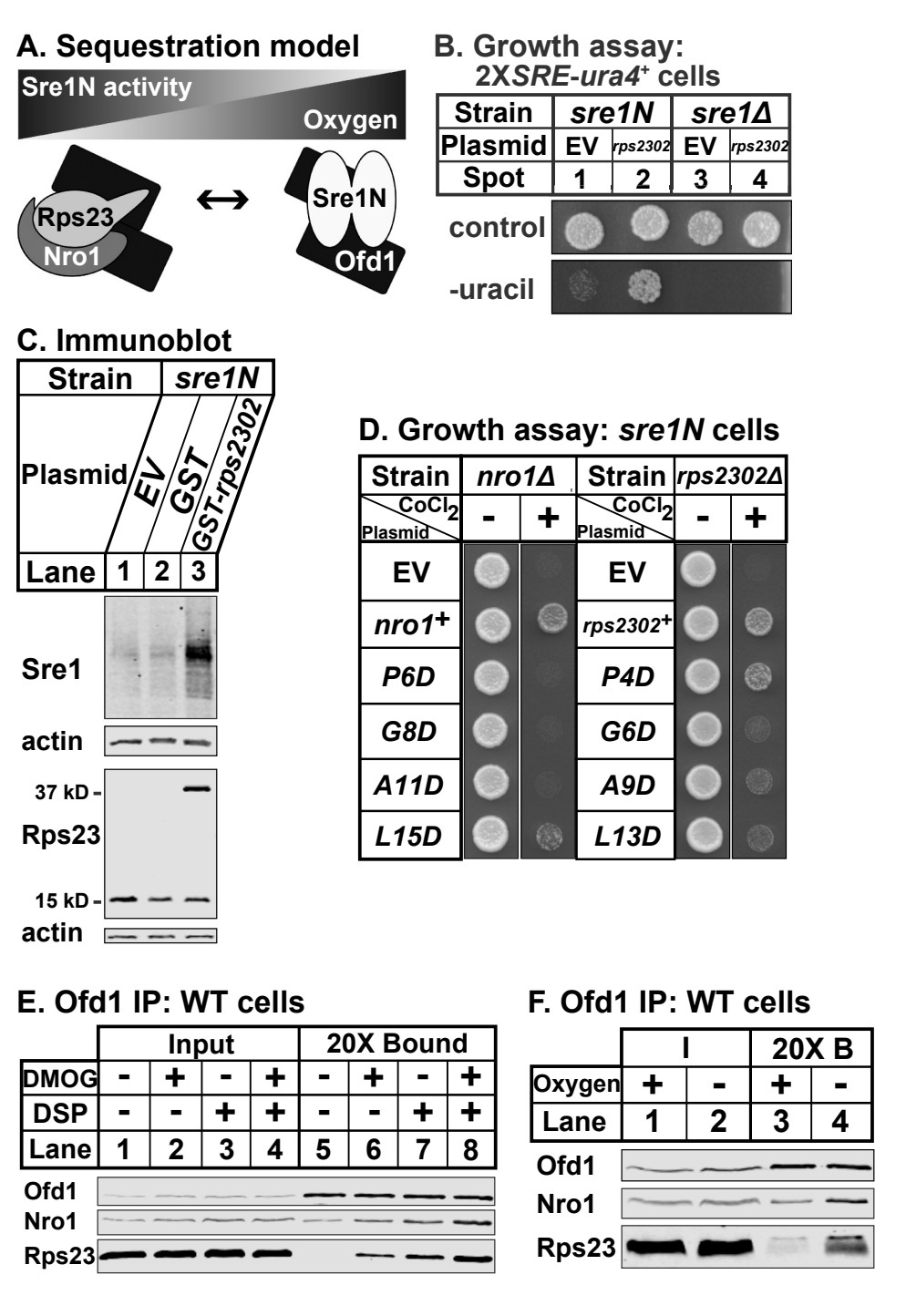

**Figure 7.** Ofd1-Rps23-Nro1 complex sequesters Ofd1 under hypoxia to activate Sre1N. (A) Sequestration model showing Ofd1 bound to either Rps23-Nro1 or Sre1N dimer. (B) *sre1N* or *sre1Δ* cells carrying an integrated *2XSRE-ura4+* reporter gene and transformed with either EV or *rps2302+* under control of the *adh1* promoter were grown on EMM-Leu or EMM-Leu-Ura media (1000 cells per spot). Shown are representative images from one of two independent experiments. (C) *sre1N* cells carrying either EV or plasmids overexpressing *GST* or *GST-rps2302* from the *nmt1* promoter were grown in minimal medium lacking thiamine for 20 hr. Whole cell extracts (100 μg) were analyzed by immunoblot using antibodies against Sre1, Rps23, and actin. Shown are representative blots from one of three independent experiments. (D) *nro1Δ sre1N* and *rps2302Δ sre1N* cells transformed with indicated plasmids under control of the CaMV and *adh* promoters, respectively, were grown (5000 cells/spot) on rich medium (3 days) and rich medium supplemented with 1.6 mM CoCl2 (4 days). Shown are representative images from one of two independent experiments. (E) Wild-type cells were cultured in rich medium in the presence of DMOG (20 mM) or DMSO (1% [v/v]) for 3 hr prior to treatment with vehicle or crosslinker. Samples were

*Figure 7 continued on next page*

*Figure 7 continued*

lysed in detergent, depleted of ribosomes, and incubated with anti-Ofd1 antibodies. Input and bound fractions were analyzed by immunoblot with indicated antibodies. Shown are representative blots from one of two independent experiments. (F) Wild-type cells were cultured in rich medium in the presence or absence of oxygen for 3 hr prior to treatment with crosslinker then processed as in (E). Shown are representative blots from one of two independent experiments.

DOI: https://doi.org/10.7554/eLife.28563.014

The following figure supplement is available for figure 7:

**Figure supplement 1.** (A) *rps23Δ::nat^r* cells were mated to *rps2302^+-kan^r* and *GST-rps2302-kan^r* cells.

DOI: https://doi.org/10.7554/eLife.28563.015

Our model also predicts that Ofd1 oxygenase activity is crucial for the oxygen-dependent sequestration of Ofd1 by Rps23-Nro1 and hypoxia stabilizes the Ofd1-Rps23-Nro1 complex due to the inability of Ofd1 to modify Rps23 P62. Ofd1 requires both 2OG and molecular oxygen as co-substrates for hydroxylation so we treated wild-type cells with dimethyloxalylglycine (DMOG), a cell permeable competitive inhibitor of Ofd1 that is metabolized to a 2OG analog, to test if enzyme activity affects Ofd1 binding to Rps23 and Nro1 (*Jaakkola et al., 2001*). Ofd1 immunopurified from cells treated with DMOG bound more Nro1 and Rps23 compared to vehicle-treated cells, in both the absence and presence of crosslinker (*Figure 7E*, lanes 5–8). Next, we examined whether Ofd1-Rps23-Nro1 complex stability is oxygen-dependent. Ofd1 immunopurified from wild-type cells bound more Rps23 and Nro1 in the absence of oxygen compared to cells grown in normoxia (*Figure 7F*, lanes 3–4). Together, these experiments show that the Ofd1-Rps23-Nro1 complex is stabilized under conditions that prevent P62 hydroxylation and support the sequestration model for Ofd1-dependent regulation of Sre1N.

## Discussion

In this study, we report three major findings: (1) Nro1 functions in a new nuclear import pathway for the small ribosomal protein uS12/Rps23; (2) Nro1 is the first protein adaptor for prolyl dihydroxylation; and (3) Rps23 has an extra-ribosomal function essential for the fission yeast hypoxic response.

Unassembled Rps23 sits at the center of these findings, and our data support the following model for how Rps23 is incorporated into ribosomes: Nro1 binds newly synthesized Rps23 in the cytosol; Ofd1 then binds Rps23-Nro1 and dihydroxylates P62 during nuclear import; Nro1 releases dihydroxylated Rps23 in the nucleus where it is incorporated into assembling 40S ribosomal subunits. Multiple lines of evidence support this model. Rps23 binds Nro1 in vitro, the interaction does not require Ofd1, and binding is stable in the absence of chemical crosslinker (*Figure 3*), suggesting that Rps23 and Nro1 bind first. The order of complex assembly is further supported by the fact that Nro1 is required for Ofd1 binding to Rps23 (*Figure 3A*), and conversely Rps23 promotes Ofd1 binding to Nro1 (*Figure 3C*). Consistent with this, efficient dihydroxylation of Rps23 requires Nro1 in vivo and in vitro (*Figure 4*). Finally, Rps23 nuclear localization requires Nro1 (*Figure 5A*), and Ofd1 nuclear import decreases in the absence of the Rps23-Nro1 complex (*Figure 5C*), indicating that import and dihydroxylation are coordinated and coincident. It remains to be determined how the complex dissociates in the nucleus and whether Ofd1-dependent hydroxylation plays a role.

Ofd1 binding to Rps23 and Nro1 is mediated through a shared binding sequence in vitro (*Figure 2*). Given previous work showing that the budding yeast enzyme crystallizes as a dimer and exists in solution as a reversible monomer-dimer mix (*Henri et al., 2010*; *Kim et al., 2010*; *Horita et al., 2015*), we modeled Ofd1 as a dimer in the complex and initially assumed that the Ofd1-binding sites in Rps23 and Nro1 contribute equally and independently to binding Ofd1 (*Figure 7A*). However, cellular studies indicate that these binding sequences are not functionally equivalent, and that sequestration of Ofd1 is more dependent on the conserved residues in Nro1 compared to Rps23 (*Figure 7D*). These findings highlight the need for future high resolution structural studies on the Ofd1-Rps23-Nro1 complex to determine both complex stoichiometry and how the Ofd1-binding sequences contribute to complex formation.

In a second key finding, Nro1 functions as an adaptor for Ofd1-dependent dihydroxylation in addition to its role as a nuclear importer. The mechanism by which Nro1 facilitates dihydroxylation of Rps23 is unclear. Previous work found that Nro1 is structurally similar to the α subunit of human

collagen prolyl-4-hydroxylase that functions in substrate recruitment (*Rispal et al., 2011*). Indeed, the study's authors predicted that Nro1 may serve a similar role for Ofd1. We report a small but significant reduction in unmodified Rps23 P62 in reactions incubated with Nro1 which supports Nro1 functioning in substrate recruitment (*Figure 4C*). However, our finding that addition of Nro1 increased production of dihydroxylated P62 at the expense of monohydroxylated P62 suggests that Nro1 facilitates the second hydroxylation event and may act by preventing release of monohydroxylated Rps23 from Ofd1. The stereochemistry of the monohydroxylated species that accumulates in *nro1Δ* cells is unknown. Previous studies showed that prolyl-3-hydroxylation is sufficient to restore the defects in translation termination caused by loss of the Ofd1 homolog Tpa1, suggesting that loss of this post-translational modification underlies the translation defects (*Loenarz et al., 2014*). Thus, we hypothesize that prolyl-3-hydroxylation of P62 is dramatically reduced in *nro1Δ* cells given the translation termination defect reported for budding yeast *ett1Δ* cells (*Henri et al., 2010*; *Rispal et al., 2011*). Since Ett1 and Nro1 are homologs, we predict that *ett1Δ* cells will show a similar loss of dihydroxylated Rps23 P62 as seen in *nro1Δ* cells. The requirement of Nro1 for Rps23 P62 hydroxylation explains why *ett1Δ* and *tpa1Δ* cells share the same read-through phenotype, while *ofd1Δ* and *nro1Δ* cells display opposing Sre1N phenotypes.

The third finding in this study is that unassembled Rps23 plays a central role in control of the hypoxic response in fission yeast. While Rps23 P62 hydroxylation in fungi is required for translation fidelity (*Loenarz et al., 2014*), we demonstrate that Rps23 has an extra-ribosomal function in the regulation of Sre1N and hypoxic gene expression. The Ofd1 hydroxylation reaction is oxygen-dependent, and this property confers oxygen regulation to the stability of the Ofd1-Rps23-Nro1 complex. The Rps23-Nro1 complex functions as a positive regulator of Sre1N by sequestering the negative regulator Ofd1 under low oxygen or DMOG-treatment (*Figure 7E–F*). We show that manipulation of unassembled Rps23 expression activates or represses Sre1N signaling in vivo (*Figures 6* and *7*). Unlike the HIF system, this sequestration mechanism allows Sre1N activity to be regulated indirectly by oxygenase activity without requiring Sre1N to be an Ofd1 substrate. While the Ofd1-Rps23 enzyme-substrate relationship is conserved in humans (*Loenarz et al., 2014*), we do not have evidence that OGFOD1 and RPS23 function in the hypoxic response in mammals, perhaps because metazoans lack an obvious Nro1 homolog.

To date, oxygen is the only known nutrient that signals to Sre1N. However, Sre1N activates anabolic pathways such as sterol synthesis that require carbon and other nutrients (*Todd et al., 2006*). Given that maintenance of cellular homeostasis requires coordination of nutrient availability and metabolism, other regulatory inputs likely exist. The unassembled ribosomal protein pool is highly dynamic and subject to multiple nutrient inputs (*Lam et al., 2007*; *Gasch et al., 2000*; *Schawalder et al., 2004*), making Rps23 an ideal regulatory molecule. We speculate that in response to nutrient availability, rates of Rps23 synthesis couple nutrient supply to Sre1N activity. Under nutrient-deprivation, reduced Rps23 synthesis dampens Sre1N activity by failing to sequester Ofd1, thereby efficiently matching lipid synthesis to nutrient supply. Conversely, high rates of Rps23 synthesis support maximal Sre1N activity by sequestering Ofd1. Cells with different levels of Rps23 synthesis still respond to hypoxia through the oxygen-dependent sequestering of Ofd1 (*Figure 6B*), but the dynamic range of this response will be tuned by nutrient supply through Rps23. In addition, Nro1 may similarly bind other Ofd1 substrates, allowing for integration of multiple signals. Current studies are focused on testing this model.

Ofd1 is a member of a large family of 2OG oxygenases that catalyze a diverse set of reactions ranging from histone demethylation for regulation of transcription to halogenation for antibiotic synthesis (*Hausinger, 2015*). Here, we describe a mechanism by which the requirement for oxygen as a substrate confers the ability to sequester the oxygenase from other binding partners under low oxygen conditions. These enzymes also require 2OG as a co-substrate, raising the possibility that 2OG can signal to Sre1N and control the hypoxic response. Finally, our studies define a new mechanism for oxygen-sensing and suggest that other 2OG-oxygenase family members may function through a similar mechanism to regulate as yet unidentified, non-substrate targets. Indeed, several 2OG oxygenases are also ribosomal oxygenases that could similarly leverage their substrates for oxygen-dependent regulation of cell function (*Ge et al., 2012*).

# Materials and methods

## Key resource table

| Reagent type (species) or resource | Designation | Source or reference | Identifiers | Additional information |
|---|---|---|---|---|
| strain, strain background (S. pombe) | WT | ATCC | | *KGY425: h- his3-D1 leu1-32 ura4-D18 ade6-M210* |
| strain, strain background (S. pombe) | *ofd1Δ* | this study | | *PEY1801: h- his3-D1 leu1-32 ura4-D18 ade6-M210 ofd1Δ::natMX6* |
| strain, strain background (S. pombe) | *ofd1 H142A D144A* | PMID: 18418381 | | *PEY1152: h- his3-D1 leu1-32 ura4-D18 ade6-M210 ofd1 H142A D144A* |
| strain, strain background (S. pombe) | *nro1Δ* | this study | | *PEY1802: h- his3-D1 leu1-32 ura4-D18 ade6-M210 nro1Δ::natMX6* |
| strain, strain background (S. pombe) | *rps2302Δ* | this study | | *PEY1803: h+ his3-D1 leu1-32 ura4-D18 ade6-M210 rps2302Δ::natMX6* |
| strain, strain background (S. pombe) | *rps23Δ; rps23Δ-natr* | this study | | *PEY1804: h- his3-D1 leu1-32 ura4-D18 ade6-M210 rps23Δ::natMX6* |
| strain, strain background (S. pombe) | *rps2302-GFP* | this study | | *PEY1805: h- his3-D1 leu1-32 ura4-D18 ade6-M210 rps2302+-GFP(S65T)-kanMX6* |
| strain, strain background (S. pombe) | *rps2302-GFP nro1Δ* | this study | | *PEY1806: h- his3-D1 leu1-32 ura4-D18 ade6-M210 rps2302+-GFP(S65T)-kanMX6 nro1Δ::natMX6* |
| strain, strain background (S. pombe) | *rps2302-GFP ofd1Δ* | this study | | *PEY1847: h- his3-D1 leu1-32 ura4-D18 ade6-M210 rps2302+-GFP(S65T)-kanMX6 ofd1Δ::natMX6* |
| strain, strain background (S. pombe) | *rps2302-GFP ofd1 H142A D144A* | this study | | *PEY1848: h- his3-D1 leu1-32 ura4-D18 ade6-M210 rps2302+-GFP(S65T)-kanMX6 ofd1 H142A D144A* |
| strain, strain background (S. pombe) | *rps2302 P62A-GFP* | this study | | *PEY1849: h+ his3-D1 leu1-32 ura4-D18 ade6-M210 rps2302 P62A-GFP(S65T)- TADH1-PURA4-kanr-TTEF* |
| strain, strain background (S. pombe) | *mCherry-ofd1* | this study | | *PEY1807: h- his3-D1 leu1-32 ura4-D18 ade6-M210 mCherry-ofd1-TADH1-PURA4-kanr-TTEF* |
| strain, strain background (S. pombe) | *mCherry-ofd1 rps23Δ* | this study | | *PEY1808: h- his3-D1 leu1-32 ura4-D18 ade6-M210 rps23Δ::natMX6 mCherry-ofd1-TADH1-PURA4-kanr-TTEF* |
| strain, strain background (S. pombe) | *mCherry-ofd1 nro1Δ* | this study | | *PEY1809: h- his3-D1 leu1-32 ura4-D18 ade6-M210 nro1Δ::kanMX6 mCherry-ofd1-TADH1-PURA4-kanr-TTEF* |
| strain, strain background (S. pombe) | *rps2302Δ sre1N* | this study | | *PEY1811: h- his3-D1 leu1-32 ura4-D18 ade6-M210 rps2302Δ::natMX6 sre1(aa1-440)* |
| strain, strain background (S. pombe) | *rps23Δ sre1N* | this study | | *PEY1812: h- his3-D1 leu1-32 ura4-D18 ade6-M210 rps23Δ::natMX6 sre1(aa1-440)* |
| strain, strain background (S. pombe) | *rps23Δ ofd1Δ sre1N* | this study | | *PEY1813: h+ his3-D1 leu1-32 ura4-D18 ade6-M210 ofd1Δ::kanMX6 rps23Δ::natMX6 sre1(aa1-440)* |
| strain, strain background (S. pombe) | *rps2302Δ ofd1Δ sre1N* | this study | | *PEY1814: h+ his3-D1 leu1-32 ura4-D18 ade6-M210 ofd1Δ::kanMX6 rps2302Δ::natMX6 sre1(aa1-440)* |

*Continued on next page*

*Continued*

| Reagent type (species) or resource | Designation | Source or reference | Identifiers | Additional information |
|---|---|---|---|---|
| strain, strain background (S. pombe) | sre1Δ | PMID: 15797383 | | PEY522: h- his3-D1 leu1-32 ura4-D18 ade6-M210 sre1Δ::kanMX6 |
| strain, strain background (S. pombe) | WT | Bioneer | | ED666: h+ leu1-32 ura4-D18 ade6-M210 |
| strain, strain background (S. pombe) | rps23Δ | Bioneer | | h+ ade6A14:E25-M210 or ade6-M216 ura4-D18 leu1-32 rps23Δ::kanMX4 |
| strain, strain background (S. pombe) | rpl23Δ | Bioneer | | h+ ade6-M210 or ade6-M216 ura4-D18 leu1-32 rpl23Δ::kanMX4 |
| strain, strain background (S. pombe) | sre1Δ | Bioneer | | h+ ade6-M210 or ade6-M216 ura4-D18 leu1-32 sre1Δ::kanMX4 |
| strain, strain background (S. pombe) | rps25Δ | Bioneer | | h+ ade6-M210 or ade6-M216 ura4-D18 leu1-32 rps25Δ::kanMX4 |
| strain, strain background (S. pombe) | GST-rps2302-kanr | this study | | PEY1816: h+ his3-D1 leu1-32 ura4-D18 ade6-M210 GST-rps2302-TADH1-PURA4-kanr-TTEF |
| strain, strain background (S. pombe) | rps2302+-kanr | this study | | PEY1817: h+ his3-D1 leu1-32 ura4-D18 ade6-M210 rps2302+-TADH1-PURA4-kanr-TTEF |
| strain, strain background (S. pombe) | ofd1Δ sre1N | PMID: 18418381 | | PEY873: h- his3-D1 leu1-32 ura4-D18 ade6-M210 ofd1Δ:: kanMX6 sre1(aa1-440) |
| strain, strain background (S. pombe) | sre1N | PMID: 18418381 | | PEY875: h- his3-D1 leu1-32 ura4-D18 ade6-M210 sre1(aa1-440) |
| strain, strain background (S. pombe) | 2XSRE-ura4+ sre1N | PMID: 15797383 | | PEY1290: h+ his-D1, leu1-32, ade6-M210, ura4-D18::2xSRE-ura4+-kanMX, sre1(aa1-440) |
| strain, strain background (S. pombe) | 2XSRE-ura4+ sre1Δ | this study | | PEY1489: h+ his-D1, leu1-32, ade6-M210, ura4-D18::2xSRE-ura4+-kanMX, sre1Δ::kanMX6 |
| strain, strain background (S. pombe) | nro1Δ sre1N | PMID: 15797383 | | PEY1410: h- his3-D1 leu1-32 ura4-D18 ade6-M210 nro1Δ:: kanMX4 sre1(aa1-440) |
| strain, strain background (S. cerevisiae) | Yeast two-hybrid | PMID: 8978031 | | AH109: MATα trp1-901 leu2-3,112 ura3-52 his3-200 gal4Δ gal80Δ LYS2::GAL1UAS-GAL1TATA-HIS3 GAL2UAS-GAL2TATA-ADE2 URA3::MEL1UAS-MELTATA-lacZ |
| recombinant DNA reagent | pSLF101 (vector) | PMID: 8332516 | | |
| recombinant DNA reagent | pART1 (vector) | PMID: 3034608 | | |
| recombinant DNA reagent | pGEX-4T3 (vector) | GE Healthcare Life Sciences | | |
| recombinant DNA reagent | pSLF172 (vector) | PMID: 9218719 | | |
| recombinant DNA reagent | pProEX-HTb (vector) | Invitrogen | | |
| recombinant DNA reagent | ppMAL-c5X (vector) | NEB | | |
| recombinant DNA reagent | EV bait (vector) | Clontech Matchmaker Gal4 Two-hybrid | Gal4_BD | |
| recombinant DNA reagent | EV prey (vector) | Clontech Matchmaker Gal4 Two-hybrid | Gal4_AD | |
| recombinant DNA reagent | human RPS23 (cDNA) | Duke Screening Center | | |

*Continued on next page*

*Continued*

| Reagent type (species) or resource | Designation | Source or reference | Identifiers | Additional information |
|---|---|---|---|---|
| recombinant DNA reagent | human OGFOD1 (cDNA) | Invitrogen Ultimate ORF | Invitrogen: 1OH23210 | |
| recombinant DNA reagent | nro1+ (plasmid) | PMID: 21481773 | | Progenitors: PCR; pSLF101 vector |
| recombinant DNA reagent | nro1 P6D, G8D, A11D, L15D (plasmids) | this study | | Progenitors: nro1+ plasmid |
| recombinant DNA reagent | rps2302+, P4D, G6D, A9D, L13D (plasmids) | this study | | Progenitors: PCR; pART1 vector |
| recombinant DNA reagent | GST-Rps23/Nro1 bacterial expression plasmids | this study | | Progenitors: PCR; pGEX-4T3 vector |
| recombinant DNA reagent | GST-rps2302 (plasmid) | this study | | Progenitors: PCR; pSLF172 vector |
| recombinant DNA reagent | GST (plasmid) | this study | | Progenitors: PCR; pSLF172 vector |
| recombinant DNA reagent | sre1N (plasmid) | PMID: 15797383 | | Progenitors: PCR; pSLF101 vector |
| recombinant DNA reagent | ofd1 bait (plasmid) | this study | | Progenitors: PCR; Gal4_BD vector |
| recombinant DNA reagent | OGFOD1 bait (plasmid) | this study | | Progenitors: OGFOD1 cDNA; Gal4_BD vector |
| recombinant DNA reagent | rps23 pombe prey (plasmid) | this study | | Progenitors: PCR; Gal4_AD vector |
| recombinant DNA reagent | RPS23 human prey (plasmid) | this study | | Progenitors:RPS23 cDNA; Gal4_AD vector |
| recombinant DNA reagent | MBP-Rps23 (plasmid) | this study | | Progenitors: PCR; pMAL-c5X vector |
| recombinant DNA reagent | ofd1 bacterial expression plasmid | PMID: 21481773 | | Progenitors: PCR; pProEX-HTb vector |
| recombinant DNA reagent | nro1 bacterial expression plasmid | PMID: 21481773 | | Progenitors: PCR; pProEX-HTb vector |
| recombinant DNA reagent | 6xHis-RPS23 (plasmid) | this study | | Progenitors: RPS23 cDNA; pProEX-HTb vector |
| antibody | IRDye800CW/IRDye680RD secondaries | LI-COR | | (1:20000) |
| antibody | anti-Nro1 (rabbit polyclonal) | PMID: 15797383 | | (1:10000) |
| antibody | anti-Dsc5 (rabbit polyclonal) | PMID: 22086920 | | (1:10000) |
| antibody | anti-Sre1 (rabbit polyclonal) | PMID: 15797383 | | (1:2500) |
| antibody | anti-Ofd1 (rabbit polyclonal) | PMID: 18418381 | | (1:10000) |
| antibody | anti-GST (mouse monoclonal) | Covance | Covance:MMS-112R | (1:1000) |
| antibody | anti-GFP (mouse monoclonal) | Roche Applied Science | Roche:1814460 | (1:1000) |
| antibody | anti-Rps23 (mouse monoclonal) | Santa Cruz | Santa Cruz:sc-100837 | (1:200) |
| antibody | anti-Rps5 (mouse monoclonal) | Santa Cruz | Santa Cruz:sc-390935 | (1:100) |
| antibody | anti-actin (mouse monoclonal) | Santa Cruz | Santa Cruz:sc-47778 | (1:200) |

Continued

| Reagent type (species) or resource | Designation | Source or reference | Identifiers | Additional information |
|---|---|---|---|---|
| Chemical compound, drug | DMOG | Frontier Scientific | Frontier Scientific:D1070 | |
| Chemical compound, drug | DSP | Thermo Fisher Pierce | Thermo Fisher Pierce:22586 | |
| Chemical compound, drug | Heavy lysine | Cambridge Isotope Laboratories | Cambridge Isotope Laboratories: CNLM-291-H-1 | |
| Chemical compound, drug | LMB | Cell Signaling | Cell Signaling:9676 | |

## Materials

Common lab reagents were obtained from either Sigma or Thermo Fisher Scientific. Oligonucleotides were provided by Integrated DNA Technologies (Coralville, IA). Heavy lysine ($^{13}C_6/^{15}N_2$) was purchased from Cambridge Isotope Laboratories (Tewksbury, MA). DMOG was from Frontier Scientific Inc (Logan, UT). Protease inhibitors (0.5 µM PMSF, 10 µg/ml leupeptin, and 5 µg/ml pepstatin) were included in buffers where indicated.

## Strains and cultures

Strains in this study are described in *Supplementary file 1* and were generated using standard techniques (*Bähler et al., 1998*; *Alfa and Cold Spring Harbor Laboratory, 1993*). All fission yeast strains were validated by PCR -sequencing and western blotting. Haploid *S. pombe* cells were cultured at 30°C in rich medium (0.5% [w/v] yeast extract (BD Biosciences, San Jose, CA), 3% [w/v] glucose, 225 µg/ml each of uracil, adenine, leucine, histidine, and lysine) to $1 \times 10^7$ cells/ml unless otherwise indicated (*Moreno et al., 1991*). Minimal medium constitutes Edinburgh Minimal Medium (MP Biomedical, Santa Ana, CA) plus supplements (225 µg/ml each of uracil, adenine, leucine, histidine, and lysine).

## Plasmids

*sre1N* (aa 1–440) under control of the constitutive CaMV promoter was described in *Hughes et al. (2005)*. *rps2302+* and *rps2302* point mutants were cloned into a constitutive *adh1*-driven expression vector with *leu2+* marker, derived from pART1 (*McLeod et al., 1987*). GST and GST-*rps2302* were placed under the control of the inducible *nmt1* promoter by insertion into the BamHI/NotI restriction sites of pSLF172 (*Forsburg and Sherman, 1997*). *nro1+* under control of the CaMV promoter was used as a template to generate *nro1* point mutants and was described in *Yeh et al. (2011)*. Transformations were performed by electroporation unless stated otherwise.

## Antibodies

Secondary antibodies were obtained from LI-COR (Lincoln, NE): IRDye800CW/IRDye680RD mouse and rabbit IgG, IRDye 680RD Detection Reagent. Commercially obtained antibodies include mouse anti-GST monoclonal (RRID:AB_291280, MMS-112R-500, Covance, Princeton, NJ), mouse anti-GFP monoclonal (RRID:AB_390913, 1814460, Roche Applied Science, Penzberg, Germany), mouse anti-Rps23 monoclonal (RRID:AB_2180354, SJ-K2, Santa Cruz, Dallas, TX), mouse anti-Rps5 monoclonal (RRID:AB_2713966, A-8, Santa Cruz), and mouse anti-actin monoclonal (RRID:AB_626632, C4, Santa Cruz). Lab-made antibodies are rabbit polyclonal antisera and were previously described: anti-Ofd1 (GST-Ofd1 full-length antigen; *Hughes and Espenshade, 2008*), anti-Nro1 (6xHis-Nro1 full-length antigen; *Lee et al., 2009*), anti-Dsc5 (6xHis-Dsc5 aa 251–427 antigen; *Stewart et al., 2012*), and anti-Sre1 (6xHis-Sre1 aa 1–260 antigen; RRID:AB_2713965, *Hughes et al., 2005*).

## Yeast two-hybrid

Yeast two-hybrid screen (>3 fold library coverage) was conducted at Duke University using a human fetal brain cDNA library. *RPS23* represented 80% of positive clones (56/70). Confirmation of screen hits was performed following Matchmaker Gal4 Two-Hybrid System user manual (Clontech, Mountain View, CA) using the yeast two-hybrid *S. cerevisiae* strain AH109 (*James et al., 1996*). Briefly,

bait (Gal4_BD empty vector, Gal4_BD-Ofd1 or Gal4_BD-OGFOD1) and prey (Gal4_AD empty vector, Gal4_AD-Rps23, or Gal4_AD-RPS23) plasmids were co-transformed into AH109 cells. Transformants were equally divided into two halves, then plated on control (SD/-Leu/-Trp) or reporter (SD/-Ade/-His/-Leu/-Trp/X-$\alpha$-Gal) plates. Plates were incubated at 30°C for 8 days and images were acquired using a flatbed scanner in transmitted light mode at a resolution of 600 dpi.

## Measuring unassembled Rps23

Strains were cultured in rich medium to 0.5–1.0 $\times$ $10^7$ cells/ml. Cells (1 $\times$ $10^8$) were pelleted and lysed by vortexing with glass beads (0.5 mm) for 10 min at 4°C in 1% [v/v] NP-40, 50 mM Hepes-HCl pH 7.4, 100 mM NaCl, 1.5 mM MgCl$_2$, and protease inhibitors. Lysates were centrifuged at 20,000 $\times$ g for 1 min at 4°C and the resulting supernatant was designated the whole cell lysate (WCL). Ribosomes were pelleted from WCL by centrifugation at 80,000 rpm for 16 min at 4°C using an Optima TLX ultracentrifuge and TLA100 rotor (Beckman Coulter, Brea, CA). WCL (5 μg) and tenfold by volume supernatant were loaded on 16% acrylamide gels. Rps23 detected in the ribosome-cleared supernatant by immunoblot was considered unassembled and normalized to Dsc5. Rps23 fractions were calculated as the mean of three biological replicates ± SEM and analyzed by paired, one-tailed $t$ test.

## GST pull-down assays

GST fusion protein expression vectors were generated by cloning gene sequences into the bacterial expression vector pGEX-4T3 (GE Healthcare Life Sciences, Chicago, IL) using BamHI/NotI restriction sites. Fusion protein expression was induced in *E. coli* BL21-CodonPlus (DE3)-RIPL competent cells (Agilent, Santa Clara, CA) with 0.1 mM IPTG at 37°C for 2 hr in LB medium. Cells were sonicated in lysis buffer (25 mM phosphate pH 6.8, 250 mM NaCl, 2 mM DTT, and protease inhibitors) prior to the addition of 1/10 vol 10% [v/v] Triton X-100. Lysates were incubated for 10 min at 4°C then centrifuged at 20,000 $\times$ g for 10 min. MagneGST particles (1.5 μl/reaction; Promega, Madison, WI) were blocked with 1% [w/v] BSA diluted in lysis buffer for 30 min at room temperature then incubated for 1 hr at room temperature with saturating amounts of cleared cell lysates. Following 3 washes in 25 mM phosphate pH 6.8, 250 mM NaCl,1% [v/v] TWEEN 20, and protease inhibitors, the particles were resuspended in 200 μl containing 84.5 nM Ofd1 or 84.5 nM Nro1 diluted in lysis buffer plus 1% [v/v] Triton X-100 for 1 hr at 4°C. Ofd1 and Nro1 were purified under native conditions as previously described (*Yeh et al., 2011*). Particles were washed twice in lysis buffer plus 1% [v/v] TWEEN 20 (no DTT) and once in 50 mM Tris-HCl pH 7.5, 100 mM NaCl prior to elution with 10 mM reduced glutathione in 50 mM Tris-HCl pH 7.5, 100 mM NaCl for 30 min at room temperature. Eluates were analyzed by immunoblotting.

## Bioinformatics

Structures were downloaded from the PDB (*Berman et al., 2003*) and analyzed using the PyMOL Molecular Graphics System (Schrödinger, LLC). Pairwise and multiple sequence alignments were generated using EMBOSS- WATER and T-coffee respectively (*Rice et al., 2000*; *Notredame et al., 2000*).

## Immunopurifications

Ofd1 and Nro1 immunopurifications (IPs) were performed as previously described (*Lee et al., 2009*) with modifications. Briefly, 1 $\times$ $10^8$ cells (Nro1 IP) or 2 $\times$ $10^8$ cells (Ofd1 IP) were pelleted, washed in PBS (137 mM NaCl, 2.7 mM KCl, 10 mM Na$_2$HPO$_4$, 2 mM KH$_2$PO$_4$, pH 7.4 plus protease inhibitors) and resuspended in 1 ml PBS plus 2 mM DSP (Pierce, Waltham, MA) or vehicle (8% DMSO unless stated otherwise) for 5 min to crosslink proteins. The reaction was quenched by addition of 1 M Tris-HCl pH 7.5 to a final concentration of 20 mM Tris. Ribosome-cleared lysates were generated as described above and 1.0–1.5 mg protein in 600 μl volume was incubated with affinity-purified antibodies (8 ng antibody/μg protein) and 30 μl Protein A agarose beads (Repligen, Waltham, MA) for 2 hr at 4°C. Beads were washed three times and resuspended in SDS lysis buffer (10 mM Tris-HCl pH 6.8, 100 mM NaCl, 1% [w/v] SDS, 1 mM EDTA, 1 mM EGTA) plus protease inhibitors prior to boiling (95°C, 5 min) and immunoblotting.

## SILAC sample preparation

Strains were cultured for $\geq 15$ generations in SILAC medium (Edinburgh minimal medium plus 75 mg/l each of leucine, histidine, adenine, uracil, and 30 mg/l heavy or light lysine) as described (Fröhlich et al., 2013). Pelleted cells were resuspended in lysis buffer (50 mM Hepes-HCl pH 7.4, 100 mM NaCl, 1% [v/v] NP-40, 1.5 mM MgCl₂) plus protease inhibitors and 1 mM DTT, and lysed by vortexing with glass beads (0.5 mm). Lysates were cleared by centrifugation at $20,000 \times$ g for 10 min and heavy-labeled or 1:1 mixtures of heavy- and light-labeled proteins were layered over a 10% sucrose cushion (20 mM Tris-HCl pH 7.2, 500 mM KCl, 5 mM MgCl₂, 10% [w/v] sucrose) plus protease inhibitors. Lysates were centrifuged at 41,000 rpm for 6 hr in the SW 41 Ti Rotor (Beckman Coulter) and ribosome pellets were solubilized in 150 µl of 6 M ultrapure urea, 3 M LiCl, 50 mM KCl, and 5 µM BME (pH adjusted to 4.5) overnight at 4°C using micro stir bars. Lysates were cleared by centrifugation at 72,000 rpm for 1 hr in the TLA100 rotor, and proteins were precipitated from the supernatant overnight with the addition of two volumes of 20% [w/v] TCA. Precipitated proteins were washed with 1:1 [v/v] ether:ethanol and air-dried prior to resuspension in 8 M ultrapure urea, 150 mM Tris-HCl pH 8.5. Samples were separated by gel electrophoresis using 16.5% acrylamide gels (Bio-Rad, Hercules, CA) and stained with Super Blue (Protea, Morgantown, WV). 15–20 kDa gel bands were excised and analyzed by LC-MS/MS.

## LC-MS/MS

Gel pieces were destained and rehydrated in 150 µl 0.01 µg/µl LysC (Wako Chemicals, Japan) in 25 mM triethylammonium bicarbonate (TEAB) then covered with an additional 150 µl of 25 mM TEAB and digested overnight at 37°C. Peptides were extracted from the gel pieces with 50% [v/v] acetonitrile, 0.1% [v/v] trifluoroacetic acid (TFA) and evaporated to dryness in a speed vac. Samples were rehydrated in 0.1% [v/v] TFA and loaded on Oasis HLB µElution solid phase extraction plates (Waters, Milford, MA) and desalted with two 100 µl aliquots of 0.1% [v/v] TFA followed by 100 µl of 10 mM TEAB. Each sample was then step fractionated under basic conditions with 10%, 25%, and 75% [v/v] acetonitrile in 10 mM TEAB yielding three fractions for each of the samples for LC/MS/MS analysis. These fractions were then dried and brought up in 10 µl of 2% [v/v] acetonitrile, 0.1% [v/v] formic acid.

The LC-MS/MS analysis was performed on a Q-Exactive HF mass spectrometer (Thermo Scientific, Waltham, MA) with a nanoACQUITY nano flow chromatography system (Waters). The samples were trapped at 5 µl/min then eluted onto a 20 cm x 75 µm i.d. C18 column for a 90 min gradient at 300 nl/min. The mass spectrometer settings were 240,000 resolution for MS and 35,000 resolution for MS2 with target of 3e6 for MS and 1e5 for MS2. Maximum injection times were set to 60 and 300 millisec respectively and a normalized collision energy of 28 was used. The data from the three fractions for each sample were combined and searched against the RefSeq2015 *Schizosaccharomyces pombe* using the Mascot (Matrix Science, London, UK) search engine running through Proteome Discoverer v.1.4 (Thermo Scientific). The precursor mass tolerance was set to 15 ppm and the fragment tolerance was set to 0.03 Da. For the search settings, the following modifications were set as variable: deamidated (N,Q), oxidation (P, M), dioxidation (P), and K8 Heavy (K). The K8 Heavy modification adds a mass of 8.014 Da, and SILAC pairs within four ppm mass were considered for calculating SILAC ratios.

## SILAC data analysis

All SILAC data were filtered for unique, high confidence peptide spectrum matches (PSMs). For quantification, missing quan values were replaced with the minimum intensity and single-peak quan channels were used. To quantify Rps23 P62 hydroxylation between SILAC pairs, first the labeling efficiency ($e$) was calculated using the median H/L of 40S protein PSMs ($j = 1, \ldots, m$) from the heavy sample alone, $s_j$:

$$e = s_{(m+1)/2} / \left( 1 + s_{(m+1)/2} \right), \text{ for ordered values of } s_j. \tag{1}$$

Next, the relative abundance of Rps23 protein in the SILAC pair was calculated using non-P62-containing H/L ratios from Rps23 PSMs ($k = 1, \ldots, q$), represented by $r_k$. These values were adjusted for labeling efficiency to give $ra_k$:

$$ra_k = abs\left[\left(\frac{1}{e}\cdot\frac{r_k}{1+r_k}\right)\middle/\left(1-\frac{1}{e}\cdot\frac{r_k}{1+r_k}\right)\right] \quad (2)$$

P62-containing peptide ratios ($p_l$) were then separated into three groups based on P62 modification state (unmodified, monohydroxylated, and dihydroxylated) and each group was analyzed independently. Ratios were normalized for Rps23 protein level to give $pn_l$ using the median of (2):

$$pn_l = p_l / ra_{(q+1)/2}, \text{ for ordered values of } ra_k. \quad (3)$$

Normalized values were adjusted for labeling efficiency as described above to give $pa_l$ and these ratios were log2 transformed. Significance was tested using Mann-Whitney, in which the $log_2(pa_l)$ values were compared against a theoretical set ($log_2(sn_j)$) where H = L calculated from $s_j$ values corrected for labeling efficiency ($sa_j$):

$$sn_j = \frac{\left(sa_j/(1+sa_j)\right)\cdot 0.5}{1-\left(sa_j/(1+sa_j)\cdot 0.5\right)} \quad (4)$$

To calculate the %H and %L for unmodified, monohydroxylated, and dihydroxylated P62, the fractional heavy ($H_l$) value of $pn_l$ was determined and corrected for labeling efficiency:

$$H_l = \frac{1}{e}\cdot\left[\frac{pn_l}{(1+pn_l)}\right] \quad (5)$$

%H and %L were reported as the median from (5) multiplied by 100, with %L equal to 100 – (%H).

To determine the relative abundance of the unmodified and monohydroxylated forms in *nro1*Δ cells, we assumed that all Rps23 P62 is unmodified in *ofd1*Δ cells. This assumption was based on the finding that over 99.9% of the dihydroxylated P62 in the wild-type-*ofd1*Δ SILAC pair originated from the heavy-labeled wild-type cells (*Figure 1F*). In addition, analysis of the *ofd1*Δ alone sample detected only unmodified Rps23 P62. The percent unmodified in *nro1*Δ cells is therefore approximated by 100 ÷ (H/L), where H/L is the median ratio of unmodified P62 in the *ofd1*Δ (H) - *nro1*Δ (L) SILAC pair. From this analysis, we found that unmodified P62 accounts for 32% of Rps23 in *nro1*Δ ribosomes. We used the same logic to calculate the percentage of dihydroxylated P62 in *nro1*Δ cells using the wild-type-*nro1*Δ SILAC pair and assuming wild-type cells contain only dihydroxylated P62. Given conservation of mass and the expectation that P62 exists in one of only three states, we then estimated the relative percentage of monohydroxylated Rps23 P62 in *nro1*Δ cells as 100 - (% unmod) – (% di), leading to the conclusion that approximately 60% of Rps23 P62 is monohydroxylated.

## MBP-Rps23 purification

*S. pombe rps23*[+] cDNA was cloned into pMAL-c5X expression vector (NEB, Ipswich, MA), and the plasmid was transformed into BL21-CodonPlus (DE3)-RIPL cells. The transformants were grown to OD600 = 0.6 at 37°C and then induced with 0.4 mM IPTG overnight at 18°C. Cells were harvested by centrifugation and cell pellets were stored at −80°C. Bacterial pellets were resuspended in lysis buffer (20 mM Tris-HCl pH 8, 150 mM NaCl) and lysed by French press. Cleared lysates were loaded onto an amylose resin column (NEB) and MBP-Rps23 was eluted with 50 mM Tris-HCl pH 7.5, 150 mM NaCl, 10 mM maltose, 1 mM EDTA, and 0.5% [v/v] Triton X-100. Fractions containing MBP-Rps23 were pooled and dialyzed in 20 mM Tris-HCl pH 8, 1 mM EDTA, and loaded onto a Mono Q anion exchanger (GE Healthcare Life Sciences). Sample was eluted with 20 mM Tris-HCl pH 8, 1 M NaCl, and purified MBP-Rps23 was collected in the flow-through, concentrated, and stored in 20 mM Tris-HCl pH 8, 150 mM NaCl.

## In vitro hydroxylation assay

For TMT analysis, reactions were performed in 20 μl volumes with Ofd1 (0.5 μM) and MBP-Rps23 (5 μM) diluted into lysis buffer (20 mM Tris HCl pH 7.5, 150 mM NaCl) with and without Nro1 (5 μM). Ofd1 and Nro1 were natively purified as previously described (*Yeh et al., 2011*). Reactions contained 20 mM Tris HCl pH 7.0, 0.1% [w/v] BSA, 0.5 mM DTT, 4 mM ascorbate, 0.15 mM FeSO$_4$, and 0.3 mM 2OG. After incubation at 37°C for 1 hr, the reaction was terminated by the addition of 1.25 μl 2 M HCl and analyzed by LC-MS/MS.

For the initial identification of hydroxylated P62, the reaction volume was scaled up to 50 µl. Ofd1 (5.2 µM) and purified human 6xHis-RPS23 (79 µM) were diluted into the lysis buffer along with 0.1% [w/v] BSA, 0.5 mM DTT, 4 mM ascorbate, 0.15 mM $FeSO_4$, and 0.3 mM 2OG. The reaction was incubated overnight at 37°C.

## TMT LC-MS/MS

Reaction samples were LysC-digested and labeled with TMT reagents (Thermo Fisher Scientific) for 1 hr at room temperature. Samples were step fractionated with four basic, reversed phase fractions of 5%, 15%, 25%, and 75% [v/v] acetonitrile in 10 mM TEAB, and run on a Q Exactive HF mass spectrometer (Thermo Scientific). The mass spectrometer settings were 120,000 resolution for MS and 60,000 resolution for MS2 over a 90 min gradient, targeting 3e6 ions for MS and 1e5 for MS2. Maximum injection times were 100 millisec for MS and 200 millisec for MS2. A stepped normalized collision energy 30/35 was used for fragmentation. The isolation window was set to 1.2 Da with a 0.5 offset, and the instrument was set to fragment the top 15 peptides by data dependent analysis with a dynamic exclusion of 10 s. An inclusion list (masses in Da: 457.95331, 458.28107, 463.28510, 463.61322, 468.61679, 468.94397) was inserted to give preference to the QPNSAIRK peptide over the highest abundance peptides. The masses in the inclusion list represented the various modification states of P62 (unmodified, monohydroxylated, and dihydroxylated), as well as the TMT label and deamidated N (+0.98 Da). To increase the signal for the dihydroxylated P62 peptide, the 75% fraction was re-run with the settings 'Do Not Pick Others.'

Data was searched against the RefSeq2015 database *Schizosaccharomyces pombe* using the Mascot (Matrix Science) search engine running through Proteome Discoverer v.1.4 (Thermo Scientific) with one missed cleavage allowed and a tolerance of 10 ppm MS and 0.03 Da MS2.

## TMT data analysis

In order to compare the relative abundance of Rps23 P62 hydroxylation in the absence and presence of Nro1, first the relative abundance of Rps23 protein between the two samples was calculated using non-P62-containing peptides. PSMs with unique QuanResultIDs were used for quantification. The intensities from each PSM ($j = 1, \ldots, m$), corresponding to the samples with and without Nro1, represented by $s_1$ and $s_2$ respectively, were summed and divided by one another to give a correction factor ($f$) for Rps23 protein levels:

$$f = \frac{\sum_{j=1}^{m} s_{1,j}}{\sum_{j=1}^{m} s_{2,j}}$$

Next, the P62-containing PSMs ($k = 1, \ldots, q$) ($p_{1,k}$ and $p_{2,k}$) were separated into three groups based on P62 modification status (unmodified, monohydroxylated, and dihydroxylated) and each group was analyzed independently. The ratios $\frac{p_{1,k}}{p_{2,k}}$ were normalized to Rps23 protein abundance and log2 transformed:

$$log_2\left(\frac{p_{1,k}}{p_{2,k}}f\right)$$

Finally, the Wilcoxon signed rank test was used to determine if the median of these ratios differed significantly from zero, the null hypothesis.

## Microscopy

Indicated strains were cultured in rich medium to $4–7 \times 10^6$ cells/ml. When indicated, cells were incubated with 92.5 nM Leptomycin B (Cell Signaling, Danvers, MA) and vehicle (0.5% [v/v] ethanol) in rich medium for 1 hr. For live-cell imaging, $5 \times 10^6$ cells were pelleted, resuspended in 15 µl rich medium, and 1 µl spotted on untreated glass slides immediately prior to imaging. Cells were imaged on a Zeiss Axio Imager M2 upright fluorescence microscope (Oberkochen, Germany). Images were captured using a Hamamatsu ORCA-ER digital camera (Japan) and iVision software (BioVision, Milpitas, California), and brightness/contrast settings were adjusted using ImageJ such that the range was identical across the strains under comparison.

## Low oxygen assays

Yeast strains were cultured and processed as previously described (*Hughes et al., 2005*) with hypoxic conditions maintained using an In vivo$_2$ 400 workstation (Biotrace, Inc, Leeds, UK). Briefly, cells grown to $1 \times 10^7$ cells/ml were resuspended in oxygenated or deoxygenated rich medium at $5 \times 10^6$ cells/ml for 3 hr then harvested and flash froze in liquid nitrogen. Cell pellets were lysed in 27 mM NaOH, 1% [v/v] 2-mercaptoethanol followed by TCA precipitation. Protein pellets were resuspended in SDS lysis buffer, and 20–100 μg protein was loaded for gel electrophoresis and immunoblotting. For alkaline phosphatase (Roche, Basel, Switzerland) treatment, 20 μg protein was diluted in 50 mM Tris-HCl pH 8.5 and incubated at 37°C for 1 hr prior to gel loading.

## Immunoblotting

Immunoblotting was performed as described in *Hughes et al. (2005)* with modifications. Unless indicated otherwise, samples were processed as described for low oxygen assays. Following electrophoresis, gels were transferred to nitrocellulose membranes using Trans-Blot Turbo transfer system (Bio-Rad). Membranes were blocked with 5% [w/v] milk in PBS-T (137 mM NaCl, 2.7 mM KCl, 10 mM Na$_2$HPO$_4$, 2 mM KH$_2$PO$_4$, 0.05%(v/v) Tween 20, pH 7.4) then incubated in primary antibody followed by an appropriate secondary (IRDye800CW or IRDye680RD mouse or rabbit IgG) and scanned using LI-COR Odyssey CLx imaging system. For immunopurification samples, IRDye 680RD Detection Reagent was added to primary antibody incubation.

## Growth assays and random spore analysis

Strains grown on rich medium agar for two days were diluted in water to $1.6 \times 10^6$ cells/ml and 3 μl were spotted on plates containing rich medium and rich medium supplemented with 1.6 mM CoCl$_2$. Plates were incubated at 30°C for the indicated time. For random spore analysis, strains were mated for two days on malt-extract plates at room temperature prior to glusulase treatment (0.5% [v/v]) for 16 hr. Spores were then diluted to $1.6 \times 10^6$ cells/ml and spotted on rich medium or rich medium plus nourseothricin (100 μg/ml) and G418 (100 μg/ml).

## RNA preparation and RT-qPCR analysis

Total RNA preparation from *S. pombe* and RT-qPCR analysis have been described previously (*Shao and Espenshade, 2014*). Briefly, total RNA was isolated using RNA STAT-60 (amsbio, Cambridge, MA). cDNA was synthesized using oligo d(T)$_{23}$VN primers (NEB). The tested genes were quantified by real-time PCR using SYBR Green qPCR master mix (Promega). *tub1$^+$* served as the internal control to calculate the relative expression across different samples. Error bars represent ± SEM of fold changes from three biological replicates.

## Human RPS23 purification

Human *RPS23* cDNA was cloned into the pProEX HTb bacterial expression vector (Invitrogen) and protein expression was induced in *E. coli* BL21-CodonPlus (DE3)-RIPL competent cells (Agilent) with 0.6 mM IPTG at 37°C for 2 hr in LB medium. Cells were pelleted and solubilized in lysis buffer (6 M Guanidine HCl, 0.5 M NaCl, and 20 mM Tris-HCl pH 8.0) (10 ml/L bacterial culture) at room temperature for 2 hr. Lysates were cleared by centrifugation in a JA20 rotor (Beckman Coulter) at 16,000 rpm for 40 min and incubated with Ni-NTA agarose (1 ml resin/10 ml lysate; Qiagen, Hilden, Germany) on a rotator for 2 hr at room temperature. Lysates plus resin were transferred to columns and washed 3x with lysis buffer. 6xHis-RPS23 was eluted 3x with lysis buffer adjusted to pH 3.5 and dialyzed overnight into 25 mM phosphate pH 6.8, 0.25 M NaCl, 2 mM DTT, and 1 mM PMSF.

## LC-MS/MS on human RPS23

In vitro hydroxylation reaction with Ofd1 and 6xHis-RPS23 was digested with 1 μg trypsin (0.2 μg/μl in 50 mM acetic acid) for 2 hr at 37°C and quenched with the addition of TFA to a final concentration of 0.1% [v/v]. Peptides were analyzed using LTQ Orbitrap Velos MS (Thermo Fisher Scientific) and parent masses (393.22 Da and 401.21 Da) were selected by SIM scan. Peptides were searched against the RefSeq 2012 database *Homo sapiens* using the Mascot V2.2.6 (Matrix Science) search engine running through Proteome Discoverer v1.3 (Thermo Scientific) with fragment match tolerance set to 0.03 Da.

## Acknowledgements

We thank the members of the Espenshade lab for their advice and acknowledge the Duke Screening Center for performing the yeast two-hybrid and Bob O'Meally from the JHMI Mass Spectrometry Core for analyzing the SILAC and TMT samples.

## Additional information

### Funding

| Funder | Grant reference number | Author |
|---|---|---|
| National Institutes of Health | HL077588 | Peter J Espenshade |
| American Heart Association | Predoctoral Fellowship 13PRE15820017 | Sara J Clasen |
| American Heart Association | Grant in Aid 13GRNT16400001 | Peter J Espenshade |
| National Institutes of Health | T32 GM007445 | Sara J Clasen |

The funders had no role in study design, data collection and interpretation, or the decision to submit the work for publication.

### Author contributions

Sara J Clasen, Conceptualization, Funding acquisition, Investigation, Writing—original draft; Wei Shao, Conceptualization, Funding acquisition, Investigation, Writing—review and editing; He Gu, Conceptualization, Investigation, Writing—review and editing; Peter J Espenshade, Conceptualization, Funding acquisition, Writing—original draft, Writing—review and editing

### Author ORCIDs

Sara J Clasen http://orcid.org/0000-0002-2888-7040
Wei Shao http://orcid.org/0000-0002-5966-4376
Peter J Espenshade http://orcid.org/0000-0002-6433-0178

### Decision letter and Author response

Decision letter https://doi.org/10.7554/eLife.28563.018
Author response https://doi.org/10.7554/eLife.28563.019

## Additional files

### Supplementary files

• Supplementary file 1. Yeast strains Genotypes and references for the yeast strains used in the study.
DOI: https://doi.org/10.7554/eLife.28563.016

• Transparent reporting form
DOI: https://doi.org/10.7554/eLife.28563.017

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
