## [Decision Letter]

Thank you for submitting your article "Coupled prolyl dihydroxylation and nuclear import of pre-ribosomal uS12/Rps23 regulate fungal hypoxic adaptation" for consideration by *eLife*. Your article has been reviewed by two peer reviewers, and the evaluation has been overseen by a Reviewing Editor and Randy Schekman as the Senior Editor. The reviewers have opted to remain anonymous.

The reviewers have discussed the reviews with one another and the Reviewing Editor has drafted this decision to help you prepare a revised submission.

Summary:

This paper describes the role of prolyl dihydroxylation of the small ribosomal subunit protein Rps23 in regulation of sterol synthesis pathways in fission yeast. It builds on a substantial body of work by the senior author on the elucidation of this pathway, in particular the identification of Ofd1 as a 2-oxoglutarate-dependent oxygenase that generates an oxygen-dependent signal leading indirectly to the degradation of the key transcription factor Sre1N.

In outline, it is now proposed that hydroxylation of Rps23 is promoted by the assembly of a protein complex involving (at least) Ofd1, Rps23 and a previously identified Ofd1-interacting protein Nro1. Binding of Ofd1 to the Nro1/Rsp23 complex is proposed to be competitive with Sre1N and modulated by the Ofd1-dependent hydroxylation of Rps23. Since binding of Ofd1 to Nro1 prevents the interaction with (and degradation of) Ser1N, production of Rsp23 and the catalytic rate for hydroxylation confer regulation on the sterol response (including by oxygen availability). This action that is proposed to be important in the co-ordination of biosynthetic processes.

The reviewers agreed that the work is of high quality and in general supports the conclusions. The manuscript should also be of interest to the wider biomedical community since Ofd1 is conserved in man as OGFOD1 and RPS23 has been identified as a prolyl hydroxylated substrate of human OGFOD1, raising questions as to the nature of signaling in human cells.

Based on this there was a consensus that the work is in principle appropriate for *eLife* but the author will need to address the following points prior to publication.

Essential revisions:

1) To strengthen Rps23 specificity in hypoxia, additional controls are suggested. In IPs with cross-linker (Figure 3) an additional negative control, similar to the one in Figure 3 should be provided. Rps5 was used to show the specificity of Nro1-Rps23 binding. Is there a reason that the authors do not present this control in other IPs with cross-linker?

2) Could the authors comment on Figure 3 with regards to no Nro1-Ofd1 binding in WT cells in Nro1 IP experiment without cross-linker and only a weak binding in the same experiment with the cross-linker (Figure 3)? How would an Ofd1 IP without cross-linker look? Does Nro1 co-enrich?

3) The authors demonstrate the crucial residues involved in interaction between Ofd1 and Rps23/Nro1 in vitro (Figure 2). However, it is important to assess the phenotypes of these mutant strains (growth) in vivo for e.g. their ability to grow on media with cobalt chloride (similarly to ofd1Δ cells). Additionally, over-expression of these Rps23 mutants in Ofd1 binding sequence should not activate expression of the reporter gene from 2XSRE-ura4+ construct. Is this the case?

4) The authors implicate a tight coupling between dihydroxylation of Rps23 and nuclear import (Figure 5). What happens under hypoxia? Is Rps23 still transported to the nucleus without the dihydroxylation?

5) Does Nro1 directly import Rps23 into the nucleus? Or does it serve as an adaptor for other import receptors? Given that Nro1 is not essential for yeast cell viability, but Rps23 is, it is important to better dissect the import pathway(s) of Rps23.

---

## [Author Response]

1) To strengthen Rps23 specificity in hypoxia, additional controls are suggested. In IPs with cross-linker (Figure 3) an additional negative control, similar to the one in Figure 3 should be provided. Rps5 was used to show the specificity of Nro1-Rps23 binding. Is there a reason that the authors do not present this control in other IPs with cross-linker?

We agree that Rps5 is an important negative control. In response to the reviewers’ suggestion, the IPs in Figure 3 that lacked the Rps5 blot (formerly A-C, now A,C,D) were repeated to include this control in place of actin.

2) Could the authors comment on Figure 3 with regards to no Nro1-Ofd1 binding in WT cells in Nro1 IP experiment without cross-linker and only a weak binding in the same experiment with the cross-linker (Figure 3)? How would an Ofd1 IP without cross-linker look? Does Nro1 co-enrich?

The new Figure 3 shows the Ofd1 IP without crosslinker (former Figure 3 moved to Figure 3). Relative to the Ofd1 IP with crosslinker, the Ofd1-Nro1 binding is weaker in the absence of crosslinker. This result is consistent with an enzyme-substrate interaction (Ofd1-Nro1/Rps23). Under both conditions, we consistently observe reduced Ofd1-Nro1 binding in *rps23Δ* and *rps2302Δ* cells. The Ofd1-Rps23 interaction is difficult to detect in the absence of crosslinker (Figure 3), as also observed in Figure 1.

3) The authors demonstrate the crucial residues involved in interaction between Ofd1 and Rps23/Nro1 in vitro (Figure 2). However, it is important to assess the phenotypes of these mutant strains (growth) in vivo for e.g. their ability to grow on media with cobalt chloride (similarly to ofd1Δ cells). Additionally, over-expression of these Rps23 mutants in Ofd1 binding sequence should not activate expression of the reporter gene from 2XSRE-ura4+ construct. Is this the case?

As suggested by the reviewers, we tested if the Ofd1-binding sequences in Nro1 and Rps23 are required to sequester Ofd1 in vivo(new Figure 7). We mutated four identical residues in the Ofd1-binding sequence to aspartate, one residue from each of the pairs of conserved residues (Figure 2). We then tested if mutant Nro1 or Rps23 rescued the cobalt growth defects of *rps2302Δ sre1N* and *nro1Δ sre1N* cells when expressed from a plasmid. If the Ofd1-binding sites in Nro1 and Rps23 contribute equally to the sequestration of Ofd1, our model predicts that the Ofd1-binding mutants will not restore growth on cobalt.

In the case of Nro1, the residues that blocked Ofd1 binding in vitro also failed to rescue growth on cobalt, consistent with the model. We verified that expression of the Nro1 mutants was similar to wild-type and directly tested Sre1N activation under +/- oxygen conditions (Figure 7—figure supplement 1). As expected, transformed cells that failed to grow on cobalt also showed impaired Sre1N signaling. Surprisingly, the analogous mutations in Rps23 had minimal impact on Sre1N levels (Figure 7—figure supplement 1) and showed partial to full rescue of growth on cobalt (Figure 7). The only Rps23 mutant that showed reduced Sre1N activation (G6D) also showed low levels of pre-ribosomal Rps23 (Figure 7—figure supplement 1). These data indicate that the two Ofd1-binding sites do not contribute equally to Ofd1 sequestration.

4) The authors implicate a tight coupling between dihydroxylation of Rps23 and nuclear import (Figure 5). What happens under hypoxia? Is Rps23 still transported to the nucleus without the dihydroxylation?

To address this concern, we expanded Figure 5 to show the Rps23-GFP localization in *ofd1Δ* and *ofd1 H142A D144A* backgrounds. Furthermore, we imaged *rps2302 P62A-GFP* cells. In these strains, Rps23 is not hydroxylated and localizes to the nucleus. Thus, import of Rps23 does not require P62 hydroxylation which is expected given the viability of *ofd1Δ* and *ofd1 H142A D144A* cells. In addition, Figure 1 shows that unmodified Rps23 assembles into functional ribosomes.

5) Does Nro1 directly import Rps23 into the nucleus? Or does it serve as an adaptor for other import receptors? Given that Nro1 is not essential for yeast cell viability, but Rps23 is, it is important to better dissect the import pathway(s) of Rps23.

This is an excellent point, and we agree that this is an interesting area for future study. Given the viability of *nro1Δ* cells, the Nro1-dependent pathway cannot be the only route for nuclear import of Rps23. Further, it remains to be tested whether Nro1 is sufficient to import Rps23. As the reviewers note, Nro1 may act as an adaptor that links Rps23 to an importin. Previous studies published by our lab identified the importin-β Kap123 as an Nro1-binding partner, and both Nro1 and Ofd1 are mislocalized in *kap123Δ* cells (Yeh et al., 2011). These data suggest that Kap123 may be an additional member of the complex. However, *rps23Δ kap123Δ* double mutants, but not *rps23Δ nro1Δ* cells, show severe growth defects, suggesting that Kap123 may separately function in Rps23 import.